

# Global CH$_4$ Fluxes Derived from JAXA/GOSAT Lower Tropospheric Partial Column Data and the CTE-CH$_4$ Atmospheric Inverse Model

Aki Tsuruta[1], Akihiko Kuze[2], Kei Shiomi[2], Fumie Kataoka[3], Nobuhiro Kikuchi[2], Tuula Aalto[1], Leif Backman[1], Ella Kivimäki[1], Maria K. Tenkanen[1], Kathryn McKain[4], Omaira E. García[5], Frank Hase[6], Rigel Kivi[1], Isamu Morino[7], Hirofumi Ohyama[7], David F. Pollard[8], Mahesh K. Sha[9], Kimberly Strong[10], Ralf Sussmann[11], Yao Te[12], Voltaire A. Velazco[13], Mihalis Vrekoussis[14,15], Thorsten Warneke[15], Minqiang Zhou[16], and Hiroshi Suto[2]

[1]Finnish Meteorological Institute, Helsinki, Finland
[2]Japan Aerospace Exploration Agency, Tsukuba-city, Ibaraki, Japan
[3]Remote Sensing Technology Center of Japan, Minato-ku, Tokyo, Japan
[4]National Oceanic and Atmospheric Administration, Global Monitoring Laboratory, Boulder, Colorado USA
[5]Izaña Atmospheric Research Center (IARC), State Meteorological Agency of Spain (AEMET), Santa Cruz de Tenerife, Spain
[6]Institute of Meteorology and Climate Research (IMK-ASF), Karlsruhe Institute of Technology (KIT), Karlsruhe, Germany
[7]Earth System Division, National Institute for Environmental Studies (NIES), Tsukuba, Ibaraki, Japan
[8]National Institute of Water & Atmospheric Research Ltd (NIWA), Lauder, New Zealand
[9]Royal Belgian Institute for Space Aeronomy (BIRA-IASB), Brussels, Belgium
[10]Department of Physics, University of Toronto, Toronto, ON, Canada
[11]Karlsruhe Institute of Technology (KIT), IMK-IFU, Garmisch-Partenkirchen, Germany
[12]Sorbonne Université, CNRS, MONARIS, UMR8233, F-75005 Paris, France
[13]Deutscher Wetterdienst (DWD), Meteorological Observatory Hohenpeissenberg, 82383 Hohenpeissenberg, Germany
[14]Climate and Atmosphere Research Center (CARE-C), The Cyprus Institute, Nicosia, Cyprus
[15]Institute of Environmental Physics, University of Bremen, Bremen, Germany
[16]Institute of Atmospheric Physics, Chinese Academy of Sciences, Beijing, China

**Correspondence:** Aki Tsuruta (aki.tsuruta@fmi.fi)

**Abstract.** Satellite-driven inversions provide valuable information about methane (CH$_4$) fluxes, but the assimilation of total column-averaged dry-air mole fractions of CH4 (XCH$_4$) has been challenging. This study explores, for the first time, the potential of the new lower tropospheric partial column (pXCH$_4$_LT) GOSAT data, retrieved by the Japan Aerospace Exploration Agency (JAXA), to constrain global and regional CH$_4$ fluxes. Using the CarbonTracker Europe-CH$_4$ atmospheric inverse

5 model, we estimated CH$_4$ fluxes between 2016–2019 by assimilating the JAXA/GOSAT pXCH$_4$_LT and XCH$_4$ data and surface CH$_4$ observations independently of each other. The Northern Hemisphere CH$_4$ fluxes derived from the JAXA/GOSAT pXCH$_4$_LT data were similar to the estimates derived from the surface observations, but was underestimated by about 35 Tg CH$_4$ year$^{-1}$ (∼6% of the global total) using the JAXA/GOSAT XCH$_4$ data. For the Southern Hemisphere, the estimates from the both GOSAT inversions were about 15–30 Tg CH$_4$ year$^{-1}$ higher than that derived from surface data. The evaluations

10 against independent data from the Atmospheric Tomography Mission aircraft campaign showed good agreement in the lower tropospheric CH$_4$ from the inversions using the JAXA/GOSAT pXCH$_4$_LT and surface data. However, the modelled North-



South gradients showed significant overestimation in the upper troposphere and stratosphere, possibly due to relatively uniform inter-hemispheric OH distributions that control $CH_4$ sinks. Overall, we found that the use of the JAXA/GOSAT pXCH$_4$_LT data shows considerable potential in constraining global and regional $CH_4$ fluxes, advancing our understanding of the $CH_4$ budget.

## 1 Introduction

Methane ($CH_4$) is the second most important greenhouse gas (GHG) after carbon dioxide ($CO_2$) with a radiative forcing of $\sim$0.650 W m$^{-2}$ (for 2022; https://www.esrl.noaa.gov/gmd/aggi/aggi.html, last access: 13 July 2024). Global and regional $CH_4$ budgets have been estimated using various data sources and methods, with recent estimates of global total emissions at 575 (553–586) Tg $CH_4$ yr$^{-1}$ over the past decade based on top-down estimates (Saunois et al., 2024). While top-down inverse models provide well-constrained global total emissions using atmospheric measurements of surface $CH_4$ and satellite total columns, regional estimates still vary significantly depending on model setups and assimilated data (Deng et al., 2024; Stavert et al., 2021).

One important factor controlling inverse model estimates is the type of data assimilated in the inverse models. Broad categories of assimilable data are 1) high-precision in-situ observations from ground-based stations, shipboard and aircraft, and 2) column-averaged dry-air mole fractions of GHGs retrieved from satellites and ground-based stations. Over the years, the column-averaged dry-air mole fractions of $CH_4$ (XCH$_4$) from various satellites, such as SCanning Imaging Absorption spectroMeter for Atmospheric CartograpHY (SCIAMACHY) on board ENVIronmental SATellite (ENVISAT) (Bovensmann et al., 1999), Thermal And Near infrared Sensor for carbon Observations–Fourier Transform Spectrometer (TANSO–FTS) on board the Greenhouse Gases Observing Satellite (GOSAT) (Kuze et al., 2009), TANSO-FTS-2 on board GOSAT-2 (Suto et al., 2021), TROPOspheric Monitoring Instrument (TROPOMI) on board the Sentinel 5 Precursor (Hu et al., 2018), have been available and used in estimation of global and regional $CH_4$ fluxes (e.g. Alexe et al., 2015; Baray et al., 2021; Chen et al., 2022; Houweling et al., 2014; Lu et al., 2021; Miller et al., 2013; Lunt et al., 2019; Pandey et al., 2016; Wang et al., 2022; Tsuruta et al., 2023; Qu et al., 2021).

Due to their spatial coverage, satellite retrievals have shown high potential in estimating GHG budgets for regions with sparse surface observations, such as the tropics (Alexe et al., 2015; Houweling et al., 2014; Qu et al., 2021), central Africa (Lunt et al., 2019; Pandey et al., 2021) and China (Chen et al., 2022; Lu et al., 2021; Wang et al., 2022). However, it has been challenging to accurately model and retrieve vertical profiles of $CH_4$ concentrations, resulting in discrepancies between XCH$_4$ estimates from transport models and satellite retrievals. For transport models, key factors influencing the estimation of XCH$_4$ include model resolution (both horizontal and vertical), estimates of tropopause height and the representation of atmospheric chemical reactions with oxidants. The latter is significant since $CH_4$ is mostly oxidized by OH (Zhao et al., 2020). For satellite retrievals, prior profiles, clouds and aerosols, surface albedo and retrieval methods contribute to the uncertainty of retrieved XCH$_4$ values (Lindqvist et al., 2024; Sha et al., 2021). These factors contribute to large-scale latitudinal and



seasonal discrepancies between the satellite retrievals and transport model estimates using prior or posterior emissions derived

by inversion estimates assimilating surface data.

Without addressing this issue, it could lead to unrealistic emission estimates from the inversions using satellite data that are significantly different from the estimates using surface observations. Previously in Tsuruta et al. (2023), we showed that the inversion using TROPOMI data without large-scale corrections could lead to smaller $CH_4$ emission estimates over the high northern latitudes compared to the inversion estimates based on surface observations. Various approaches have been developed

to manipulate the large-scale discrepancies, which include: adjusting large-scale discrepancies before performing satellite inversions (Houweling et al., 2017; Wang et al., 2022; Zhang et al., 2021), using the so-called proxy method that optimizes the $CO_2$:$CH_4$ ratios based on GOSAT data that provide both $XCO_2$ and $XCH_4$ retrievals (Feng et al., 2017; Palmer et al., 2021; Pandey et al., 2016), or discarding high-latitude data, where the problem appears most severe (Alexe et al., 2015; Baray et al., 2021; Lu et al., 2022). Apart from the proxy method, these adjustments have been somewhat arbitrary, with the degree

of adjustments varying between studies. With appropriate manipulations, results from inversion using surface and satellite data seem to agree in general, while satellite data can also provide additional regional information about magnitude and seasonality of $CH_4$ emissions (e.g. Wang et al., 2022; Lu et al., 2021; Pandey et al., 2021; Lunt et al., 2019; Feng et al., 2017).

The TANSO-FTS has measured reflected sunlight with two orthogonal components of polarization in the short wave infrared (SWIR) and thermal emissions in the thermal infrared (TIR) simultaneously at the local time of 13:00. SWIR data constrain

the total column density and TIR data provide vertical profile information. Recently, the Japan Aerospace Exploration Agency (JAXA) has developed a new retrieval product of the partial column $CO_2$ and $CH_4$ densities in the lower troposphere (LT, typically 0–4 km), upper troposphere (UT, typically 4–12 km) and stratosphere from the SWIR and TIR by minimizing contamination by highly-polarized radiation scattered by aerosols and thin clouds (Kikuchi et al., 2016; Kuze et al., 2022). Compared to total column retrievals, this method offers the advantage that lower and upper tropospheric products contain more information

about surface fluxes, making them particularly useful for detecting local $CH_4$ (Kuze et al., 2020) and $CO_2$ (Kuze et al., 2022) fluxes. Atmospheric transport models generally perform well representing the in lower troposphere and combined with inverse models, they can reproduce atmospheric $CH_4$ surface observations reasonably well. Therefore, the use of tropospheric partial column data may provide better constrains for global and regional $CH_4$ flux estimates than using total column data.

In this study, we present for the fist time a way to assimilate JAXA/GOSAT lower tropospheric partial column $CH_4$

(pXCH4_LT) data into the atmospheric inverse model CarbonTracker Europe-$CH_4$ (CTE-$CH_4$; Tsuruta et al., 2017). We examined the global $CH_4$ fluxes for 2016–2019 derived from the pXCH4_LT data and compare those to the flux estimates derived from JAXA/GOSAT $XCH_4$ data and surface $CH_4$ observations. We evaluated annual budgets, seasonal cycles and spatial distributions of the total and subcategory emissions (anthropogenic and wetlands). Additionally, we compared optimized atmospheric $CH_4$ to independent (i.e. not assimilated) data from the Atmospheric Tomography Mission (ATom) aircraft cam-

paign and total and lower tropospheric partial column data from Total Carbon Column Observing Network (TCCON). The study highlights the potential of JAXA/GOSAT pXCH4_LT data to improve the constraints on global and regional $CH_4$ fluxes compared to total column data.





## 2 Method

### 2.1 CTE-CH$_4$

CarbonTracker Europe-CH$_4$ (CTE-CH$_4$; Tsuruta et al., 2017) is a modular atmospheric inverse modelling system (van der
Laan-Luijkx et al., 2017) based on the ensemble Kalman filter (Peters et al., 2005). It minimizes the cost function $J$,

$$J = (\mathbf{x} - \mathbf{x^b})\mathbf{P^{-1}}(\mathbf{x} - \mathbf{x^b}) + (\mathbf{y} - \mathcal{H}(\mathbf{x^b}))\mathbf{R^{-1}}(\mathbf{y} - \mathcal{H}(\mathbf{x^b})), \qquad (1)$$

where $\mathbf{x}$ is the state vector, $\mathbf{P}$ is the state covariance matrix, $y$ is the observation of atmospheric concentrations (see Section
2.3), $\mathcal{H}$ is the observation operator, and $\mathbf{R}$ is the observation covariance matrix. In this study, the state vector $x$ included the

flux multiplication factors for anthropogenic and wetland fluxes (see Section 2.2). Fluxes were optimized at $1° \times 1°$ (latitude
$\times$ longitude) resolution for land areas in northern Eurasia, $2° \times 3°$ grid for other land areas, and region-wise over the ocean
(Fig. A1). Note that we do not optimize natural ocean fluxes, but do optimize anthropogenic emissions over oceans, such as
shipping and flight tracks that were included in the prior fluxes (EDGAR v8.0, Section 2.2). The prior covariance $P$ was a
block diagonal matrix, assuming that $1° \times 1°$ optimization regions were uncorrelated with $2° \times 3°$ optimization regions, land

and ocean regions were uncorrelated and wetland fluxes were uncorrelated with anthropogenic fluxes. The prior uncertainty
(diagonal values) were defined as ratio to prior fluxes (Section 2.2): 80% over land and 20% over ocean. Off-diagonals were
defined based on distances between the grids and regions with spatial correlation lengths of 100 km for $1° \times 1°$ optimization
regions, 300 km over $2° \times 3°$ optimization regions and 900 km over the ocean. Localization schemes as in Peters et al. (2007)
were applied.

For the observation operator $\mathcal{H}$, the atmospheric transport model TM5 (Krol et al., 2005) was used. TM5 was run at $1° \times 1°$
over Europe and $6° \times 4°$ globally, constrained by 3-hourly European Centre for Medium-Range Weather Forecasts (ECMWF)
ERA5 meteorology (Hersbach et al., 2020). Atmospheric chemistry included OH, Cl and O($^1$D) as in Houweling et al. (2014),
where the reaction rates were calculated from the ECHAM/MESSy1 model (Jöckel et al., 2006). The initial CH$_4$ concentration
fields were taken from Tenkanen et al. (2024).

### 2.2 Prior fluxes

For anthropogenic emissions, prior estimates were taken from the Emissions Database for Global Atmospheric Research
(EDGAR) v8.0 (Crippa et al., 2023). For fluxes from wetlands and dry mineral soils (hereafter wetlands), the estimates from
the LPX-Bern v1.4 process based ecosystem model (Lienert and Joos, 2018) were used with 2019 values replicated from 2018.
Other sources include biomass burning, microbial (termites) and ocean emissions, and the estimates from Global Fire Emis-

sions Database (GFED) v4.1s (van der Werf et al., 2017), the Vegetation Integrative SImulator for Trace gases (VISIT; Ito
and Inatomi, 2012) and Saunois et al. (2020) were used, respectively. Among those, the fluxes from anthropogenic, wetlands
and biomass burning were monthly and inter-annually varying. The emissions from termites were annual estimates, and ocean
fluxes were climatological.



## 2.3 Atmospheric observations

### 2.3.1 JAXA/GOSAT partial column data

JAXA/GOSAT retrieval algorithm is based on Full Physics algorithm extended simultaneous use both 2-orthogonal SWIR signal and TIR. The algorithm is based on Kikuchi et al. (2017). JAXA's forward calculation constructs the vector radiance and use 2-orthogonal polarized SWIR observation in four windows with bi-directional reflection. For TIR, the scalar radiance is handled in forward model for three windows. The an empirical orthogonal function (EOF) fitting are taken account in retrieval process. In the retrieval process, $XCO_2$, $XCH_4$, and $XH_2O$ are simultaneously retrieved with solar-induced chlorophyll fluorescence (SIF) information. Two layers in the tropospheric and three layers in the stratospheric $CO_2$ and $CH_4$ partial column-averaged concentrations are derived. For $H_2O$, 11 layers of vertical concentration are derived.

The five layers are defined by pressure levels based on the retrieved surface pressure, denoted as $sp_{GOSAT}^{ret}$. The lower and upper troposphere are defined as between $sp_{GOSAT}^{ret}$ and $0.6 \times sp_{GOSAT}^{ret}$ and between $0.6 \times sp_{GOSAT}^{ret}$ and $0.2 \times sp_{GOSAT}^{ret}$ pressure levels, respectively. For three stratospheric layers, between $0.2 \times sp_{GOSAT}^{ret}$ and $0.1 \times sp_{GOSAT}^{ret}$, between $0.1 \times sp_{GOSAT}^{ret}$ and $0.05 \times sp_{GOSAT}^{ret}$, and between $0.05 \times sp_{GOSAT}^{ret}$ and $0.01$ are defined as lower stratosphere, middle stratosphere, and upper stratosphere, respectively. In this study, all the GOSAT data is based on JAXA/GOSAT product (v2.0). (https://www.eorc.jaxa.jp/GOSAT/Global_GHGs_Map/index.html, last access: 4 June 2024).

During the study period, TANSO-FTS and Cloud and Aerosol Imager (CAI) were shut down and the observation had been suspended during 17–24 May 2018 due to the Command and Data Management System (CDMS) incident, and from 24 November 2018 until 28 December 2018 due to the rotation anomaly of the second solar-paddle.

For comparison to model estimates, the $XCH_4$ values from model results were calculated as:

$$XCH_4^{model} = \frac{\sum_i (CH_{4i} \times dp_i)}{\sum_i dp_i}, \quad i = 1, ..., 25 \tag{2}$$

where $CH_{4i}$ is the dry air mixing ratio of $CH_4$ at TM5 model level $i$, temporally and horizontally interpolated to time and location of the GOSAT data, and $dp_i$ is the pressure thickness at the level $i$.

For calculating modelled partial columns of methane, $pXCH_4\_LT^{model}$, the levels were selected such that the minimum level $i$ corresponded to the index where the GOSAT-retrieved surface pressure exceeded the TM5 model layer pressure, and the maximum level corresponded to the point where GOSAT pressure reached $sp_{GOSAT}^{ret} \times 0.6$. Vertical interpolation was not applied, leading to potential biases.

Averaging kernels were not applied to model $XCH_4^{model}$ and $pXCH_4\_LT^{model}$, as it was unavailable for the v2.0 product. However, we are aware of a newer version, v3.0 data, where averaging kernels are now available. We did not apply any preprocessing of the JAXA/GOSAT data, such as averaging or removing large-scale differences that may have been raised in comparison to inversion estimates assimilating surface data. This way, we can examine the effect of observations directly. In the assimilation, we assumed observational uncertainty (retrieval error + transport model error) to be 1) 30 ppb globally with a rejection threshold of 60 ppb in both cases assimilating $XCH_4$ or $pXCH_4\_LT$, and 2) 50 ppb with a rejection threshold of 100 ppb and assimilating only $pXCH_4\_LT$ data over land (see also 2.4). These values are somewhat arbitrary, but following other



inversion experiments using the GOSAT data (e.g. Janardanan et al., 2020; Lu et al., 2021; Maasakkers et al., 2021; McNorton et al., 2018). The larger uncertainty is examined as retrieval errors in the lower tropospheric partial column data are probably higher compare to the total column data.

### 2.3.2 Surface CH$_4$ mole fractions

Surface CH$_4$ mole fraction observations mainly from ObsPack v4.0 (Schuldt et al., 2021) were assimilated in the inversion, as well as used for evaluation (see Fig. A1 for site locations and Table A1 for details). The data include ICOS ATC CH$_4$ Release and ICOS ATC NRT CH$_4$ growing time series data, which were downloaded among with other NOAA ObsPack data and the ICOS Carbon portal (Table A2). The data consisted of discrete and continuous observations from in-situ stations and ships. Similar to our previous studies (Tsuruta et al., 2017, 2019; Tenkanen et al., 2024), all data were filtered by taking observations at well-mixed conditions based on quality flags given by the data providers. Continuous data were processed into daily means by averaging observations during local time afternoon (12–16) or night (0–4). Observational uncertainties (observational error + transport model error) ranged between 4.5–75 ppb, depending on each site (Table A1).

In addition, we used aircraft measurements from the Atmospheric Tomography Mission (ATom; Thompson et al., 2022), obtained from the ObsPack v6.0 (Schuldt et al., 2023) for independent evaluation. During 2016–2019, there were ATom observations from four campaigns: 2016 July–August, 2017 January–February, September–October and 2018 April–May. Prior and posterior mole fractions were estimated at each sampling location and time, linearly interpolated within the TM5 model grid cells. ATom data were not assimilated in the inversions and therefore can be used as independent observations for validation.

### 2.3.3 TCCON

The Total Carbon Column Observing Network (TCCON) is a global network providing XCH$_4$ measurements retrieved from the spectrum of near-infrared radiation of direct sunlight using ground-based Fourier Transform Spectrometers (FTS) (Wunch et al., 2011). We used the GGG2020 data (Laughner et al., 2023, 2024) from 25 sites globally (Table A3) for evaluation of inversion results. The sites were selected as those that provided GGG2020 data, and have at least one year of measurements between 2016–2019. The data were not assimilated in the inversions and so can also be used as independent observations for validation.

For comparison, hourly average mixing ratio interpolated horizontally to the TCCON locations were used. Temporal co-locations were done by selecting the TCCON observations that were closest to the model time step (hourly) and set the time limit of half an hour; if there was a TCCON observation made within ± half an hour of the model time step, the TCCON and modelled values were taken into account.

For comparison to total column (XCH$_4$), the model estimates were calculated by applying TCCON averaging kernels (Rodgers and Connor, 2003):

$$\hat{c} = c_a + (\mathbf{h} \circ \mathbf{a})^{\mathbf{T}}(\mathbf{x} - \mathbf{x_a}) \tag{3}$$



where $\hat{c}$ is the averaging kernel corrected $XCH_4$ value from the model, $c_a$ is the TCCON prior $XCH_4$, $\mathbf{h}$ is the TCCON pressure weighting function, $\mathbf{a}$ is the TCCON averaging kernel, $\mathbf{x}$ is the model profile and $\mathbf{x_a}$ is the TCCON prior profile. After applying
the averaging kernel, daily means were calculated for evaluation.

     In addition, the tropospheric partial column were calculated from TCCON total columns of $CH_4$ and hydrogen fluoride (HF). Practically all of the HF exists in the stratosphere, where HF is produced from photodissociation of chlorofluorocarbons (CFCs). The concentrations of long lived tracers, such as $CH_4$ and HF, are strongly correlated in the stratosphere (e.g. Plumb, 2007). Airmasses containing both $CH_4$ and CFCs enter through the topical stratosphere, where $CH_4$ is oxidized and HF is
produced. In the stratosphere, $CH_4$ shows a nearly linear inverse relationship with HF. By assuming a linear relationship in the stratosphere (Washenfelder et al., 2003; Saad et al., 2014) the stratospheric partial column of $XCH_{4\,TCCON}^{tropo}$ is given as

$$XCH_{4\,TCCON}^{tropo} = XCH_4 - \beta \times XHF \tag{4}$$

where $\beta$ is the stratospheric $CH_4$:HF slope. As the tracer to tracer correlations typically exhibit distinct correlations in the tropics, extratropics and polar stratospheric vortices (Plumb, 2007), we derived slopes for the tropics (30° S–30° N), SH (90°
S–30° S), NH (30° N–90° N), as well as for the polar stratospheric vortices. Profile data of $CH_4$ and HF from the Atmospheric Chemistry Experiment - Fourier Transform Spectrometer (ACE-FTS) version 4.1 data products (Boone et al., 2020) were used to calculate the slopes. The polar vortices were identified from potential vorticity data from the ERA5 reanalysis (Hersbach et al., 2020). The vortex edge was assumed to be represented by the $|36|$ PVU isoline. The interference of water can increase the error in the column averaged HF (XHF) (Saad et al., 2014), especially at high airmasses. Therefore, observations with
column averaged $H_2O$ ($xH_2O$) above 2000 ppm and zenith angle larger than $90 - 0.0075 \times xH_2O$ degrees were discarded.

     For comparison to the tropospheric partial column, Eq. 2 was applied similarly to calculate $pXCH_4\_LT^{model}$, but for the lowest and uppermost layer, vertical interpolation was applied to TCCON retrieval surface pressure $sp_{TCCON}^{ret}$ and tropopause height, respectively. The pressure at dynamic tropopause was calculated based on ERA5 reanalysis data, defined by the $|2|$ PVU isoline. Constant extrapolation was applied in case $sp_{TCCON}^{ret} > sp^{model}$.

## 2.4   Simulation setups

In this study, we present the results from four inversions that differed in the observations assimilated:

     InvSURF: surface $CH_4$ data only

     InvGLT: JAXA/GOSAT partial column data ($pXCH_4\_LT$) only

     InvGLT_land: JAXA/GOSAT partial column data ($pXCH_4\_LT$) over land only
InvGTOT: JAXA/GOSAT total column data ($XCH_4$) only

In all of the GOSAT inversions, only the JAXA/GOSAT data were assimilated to examine the effect of the data in constraining fluxes independent of other datasets. For InvGLT and InvGTOT, observational uncertainty and rejection threshold was set to be 30 and 60 ppb, respectively, and for InvGLT_land they were set as 50 and 100 ppb, respectively. All inversions were run for 2015–2019, but 2015 was considered as a spin-up, and not included in the analysis. The analysis on regional fluxes was done





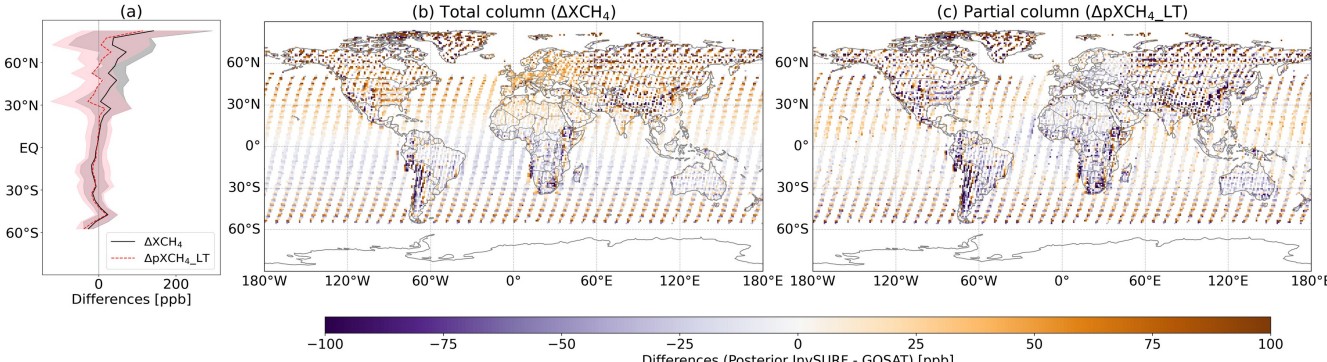

**Figure 1.** Mean differences between posterior InvSURF and JAXA/GOSAT retrievals, averaged over 2016–2019. (a) five degree latitudinal means (solid line) with standard deviation (shaded area), and (b) and (c) $1° \times 1°$ grid means. Positive values indicate posterior InvSURF mole fractions being higher than the JAXA/GOSAT retrievals.

based on 30° latitudinal bands. Throughout the following sections, the prior and posterior mole fractions refer to those derived using prior and posterior fluxes, respectively.

## 3 Results

### 3.1 Comparison of CH$_4$ mole fractions from InvSURF and JAXA/GOSAT

In this section, we analyse the differences between posterior estimates from the surface-based inversion (InvSURF) and JAX-

210 A/GOSAT retrievals for total and partial columns, i.e.

$$\Delta XCH_4 = XCH_{4}{}^{post}_{InvSURF} - XCH_{4}{}^{ret}_{GOSAT}$$
$$\Delta pXCH_4\_LT = pXCH_4\_LT^{post}_{InvSURF} - pXCH_4\_LT^{ret}_{GOSAT} \quad (5)$$

In the comparison of total column, latitudinal biases were found with positive $\Delta XCH_4$ values in the extratropics and negative values in the tropics, especially in the SH tropics (Fig. 1 (b)). This feature was systematic in time, such that similar biases were found regardless of years and seasons, although the absolute value of the biases varied. In the comparison of lower tropospheric

partial columns, such latitudinal biases were less prominent (Fig. 1 (c)), especially over land, indicating better agreement over NH extratropics (Fig. 1 (a)). This indicates the potential role of upper atmosphere in the total column biases, where posterior estimates and retrievals had difficulty agreeing.

Large biases were observed in regions such as Greenland, western South America, southernmost South Africa, eastern China and northern Russia in both comparisons (Fig. 1 (b) and (c)). Because of the challenges in retrieving data over ice-covered

land, we assume that biases in Greenland were mostly associated with retrieval errors. Biases in other regions could be due to unresolved fluxes by surface observations, i.e. the inversion error in estimating fluxes, retrieval errors due to cloud cover and





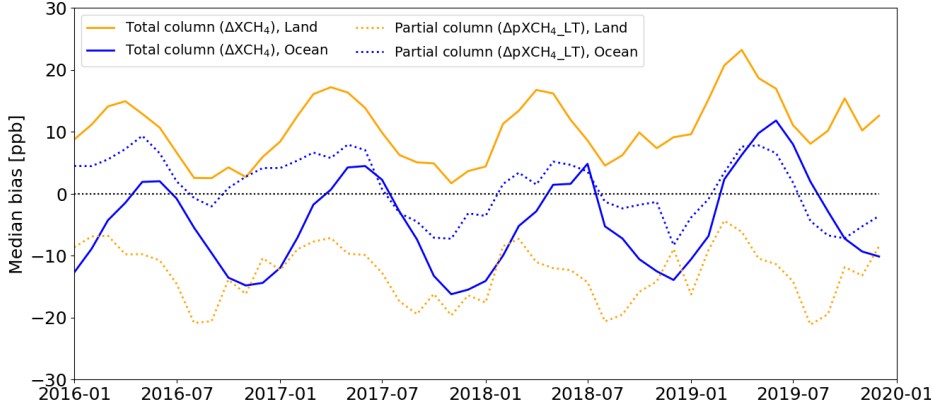

**Figure 2.** Global monthly median differences between posterior InvSURF and JAXA/GOSAT retrievals. Positive values indicate posterior InvSURF mole fractions being higher than the JAXA/GOSAT retrievals.

difficulties in retrieving surface pressure in regions with highly elevated surface. Europe showed the smallest $\Delta pXCH_4\_LT$, which is encouraging considering that InvSURF probably had constrained the emission in Europe the best compared to other regions globally, and modelled tropospheric $CH_4$ from InvSURF is well in line with ground-based observations (Section 3.2).

Both the transport model and optimization resolutions were the smallest and number of observations were relatively large in Europe (Section 2.1). There was a notable shift between better agreement in Europe and a worse agreement in Russia and Africa, attributed to the CTE-CH$_4$ setup: TM5 was run with highest spatial resolution over Europe and an extended area (Tsuruta et al., 2015), with a coarser resolution elsewhere. This resulted to creating an artifact of a border between the zoom grid and the global grid that caused the shift. An additional simulation with TM5 with global $2° \times 3°$ (latitude $\times$ longitude)

resolution and without zoom, eliminated these boundaries (Fig. A2).

Both lower tropospheric partial and total column comparisons showed seasonal and land-sea biases (Fig. 2). The global average $\Delta XCH_4$ and $\Delta pXCH_4\_LT$ were smaller during NH winter than summer. The amplitude of the seasonal biases was larger in the total column, especially over oceans. The comparison of the land and sea biases it showed that the biases in the total column were larger over land than ocean, whereas an opposite behaviour was observed for the lower tropospheric

partial columns. In other words, the $\Delta XCH_4$ and $\Delta pXCH_4\_LT$ were closer over ocean than land on average, with absolute median monthly differences of 5 ppb over ocean and 23 ppb over land, indicating the possible influence of land fluxes to upper atmosphere. Overall, the average bias during 2016–2019 was smallest in $\Delta pXCH_4\_LT$ over ocean.

## 3.2 Evaluation against surface and aircraft data



**Table 1.** Bias, root mean squared error (RSME) and Pearson's correlation against observations at surface ground-based stations assimilated in InvSURF, ATom aircraft measurements, and TCCON data. The statistics were calculated for each ground-based and TCCON station over 2016–2019 and four ATom campaigns separately, and the mean of all stations or campaigns are shown. The value followed by ± sign is the standard deviation (std) of statistics from all stations. For ATom, the minima and maxima of the statistics are shown in square brackets instead of std due to the limited number of campaigns (four) to calculate std from. Biases were calculated as modelled values subtracted by observed values, and therefore, positive values indicate model overestimation. The simulation with the best statistics are highlighted in bold.

| *Observations*/Inversions | Bias [ppb] | RMSE [ppb] | Correlation |
|---|---|---|---|
| *Surface ground-based* | | | |
| Prior | -42.2 ± 15.6 | 20.3 ± 11.1 | 0.69 ± 0.15 |
| InvSURF | **-9.2 ± 10.7** | **18.1 ± 10.5** | **0.80 ± 0.12** |
| InvGLT | -12.3 ± 13.6 | 20.8 ± 13.2 | 0.74 ± 0.15 |
| InvGLT_land | -11.7 ± 13.8 | 20.6 ± 13.1 | 0.74 ± 0.15 |
| InvGTOT | -36.9 ± 18.8 | 20.0 ± 12.1 | 0.74 ± 0.16 |
| *ATom, < 2000 m* | | | |
| Prior | -35.2 [-40.6, -31.1] | 16.0 [11.0, 22.2] | 0.97 [0.93, 0.98] |
| InvSURF | -13.4 [-20.8, -8.7] | **15.8** [12.2, 18.0] | **0.97 [0.95, 0.98]** |
| InvGLT | -11.1 [-19.0, -5.1] | 18.3 [14.3, 25.9] | 0.95 [0.91, 0.97] |
| InvGLT_land | **-8.5** [-15.9, -4.0] | 19.9 [15.4, 26.3] | 0.94 [0.90, 0.97] |
| InvGTOT | -28.6 [-37.4, -21.9] | 22.0 [15.9, 34.9] | 0.94 [0.87, 0.98] |
| *ATom, all* | | | |
| Prior | -22.8 [-34.3, -15.6] | **34.6** [17.3, 48.9] | 0.77 [0.63, 0.92] |
| InvSURF | 3.3 [-11.2, 8.1] | 35.2 [18.1, 49.9] | 0.77 [0.63, 0.92] |
| InvGLT | **1.2** [-8.3, 13.6] | 34.7 [17.5, 48.1] | 0.76 [0.64, 0.92] |
| InvGLT_land | 3.5 [-5.4, 16.0] | **34.6** [18.8, 47.7] | 0.76 [0.65, 0.90] |
| InvGTOT | -13.8 [-26.0, -0.7] | 34.9 [19.2, 45.8] | **0.76 [0.66, 0.92]** |
| *TCCON, partial column* | | | |
| Prior | -3.3 ± 16 | 15.4 ± 4.0 | 0.38 ± 0.20 |
| InvSURF | 23.4 ± 20.3 | **13.6 ± 4.2** | **0.68 ± 0.15** |
| InvGLT | 24.2 ± 15.8 | 15.8 ± 5.5 | 0.56 ± 0.21 |
| InvGLT_land | 25.4 ± 15.4 | 15.7 ± 5.4 | 0.56 ± 0.20 |
| InvGTOT | **0.7 ± 9.0** | 14.7 ± 4.5 | 0.59 ± 0.19 |
| *TCCON, total column* | | | |
| Prior | **3.2 ± 15.3** | 12.4 ± 7.9 | 0.39 ± 0.23 |
| InvSURF | 23.8 ± 20.4 | **10.1 ± 6.4** | **0.59 ± 0.28** |
| InvGLT | 25.2 ± 18.4 | 11.9 ± 8.8 | 0.50 ± 0.28 |
| InvGLT_land | 26.3 ± 18.3 | 11.8 ± 8.7 | 0.50 ± 0.27 |
| InvGTOT | 8.1 ± 10.8 | 11.2 ± 7.4 | 0.52 ± 0.28 |



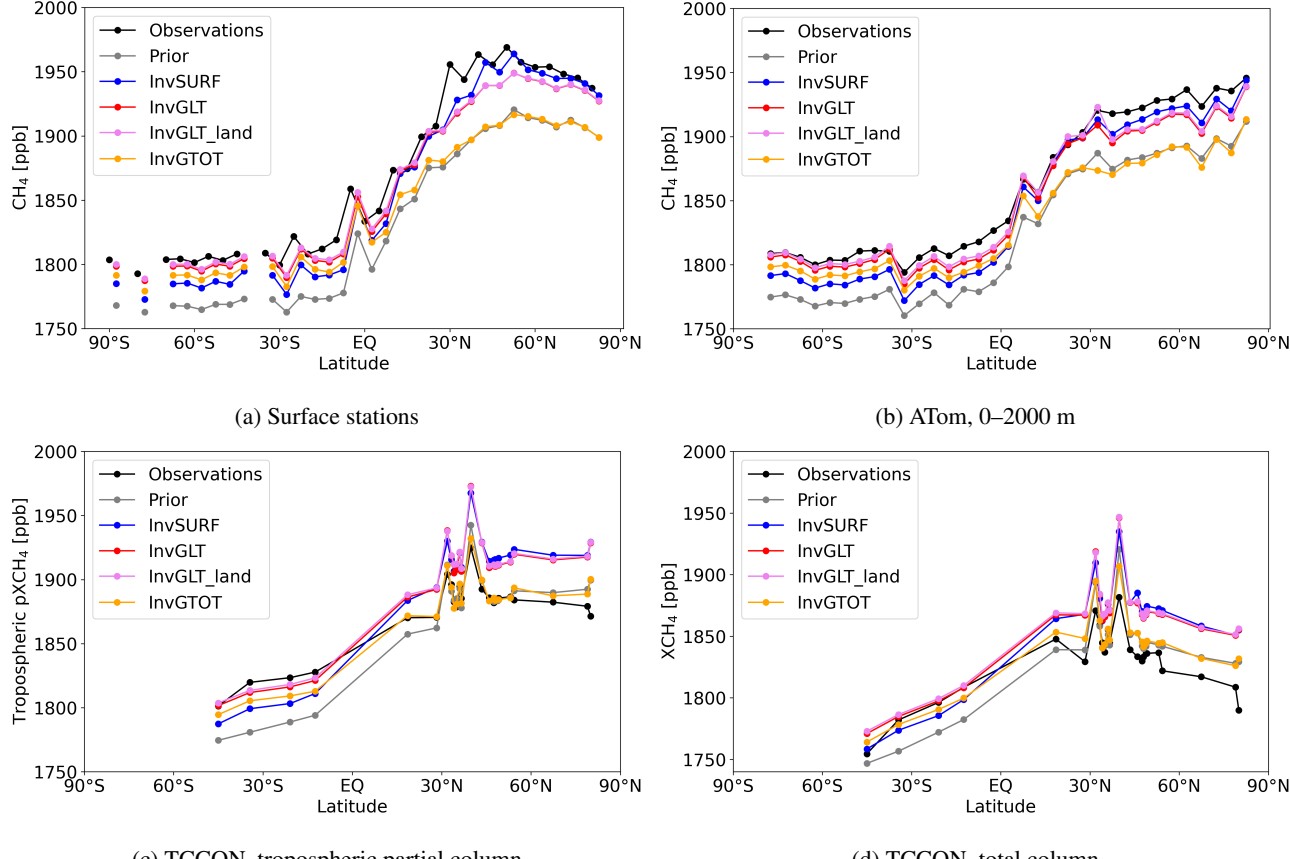

(a) Surface stations

(b) ATom, 0–2000 m

(c) TCCON, tropospheric partial column

(d) TCCON, total column

**Figure 3.** Mean atmospheric CH$_4$, averaged over 2016–2019 at (a) surface ground-based and shipboard stations, (b) ATom observation locations, and (c) and (d) TCCON sites. For (a), the data that were assimilated in InvSURF were used. For (a) and (b), mean over five-degree latitude bands are shown. For (c) and (d), no spatial averaging is applied.

Comparison of posterior atmospheric CH$_4$ to the surface ground-based observations showed the smallest overall bias, root
mean squared error (RMSE) and strongest correlation for InvSURF (Table 1) as expected, since these observations were as-
similated in the inversion. Among GOSAT inversions, those using lower-tropospheric data (InvGLT and InvGLT_land) showed
the best agreement to the surface ground-based observations, while the total column inversion (InvGTOT) showed large neg-
ative biases following the prior (Table 1). RMSE and correlation of the GOSAT inversions were not significantly different.
The latitudinal gradient was best captured by InvSURF, InvGLT and InvGLT_land compared to the surface stations (Fig. 3a).
The inversions mostly underestimated the surface observations with the underestimation being the smallest in the Southern
Hemisphere (SH) for InvGLT and InvGLT_land and in the Northern Hemisphere (NH) for InvSURF. InvGTOT showed better
agreement than InvSURF in the SH, but considerable underestimation in the NH (Fig. 3a), which was the reason for strong
negative bias in the overall agreement (Table 1).





In the mid- and high northern latitudes (above 30° N) the seasonal cycle amplitude (SCA) was generally overestimated by model estimates in prior, which was worsened by inversion (SCA of posterior estimates were larger than prior) (Fig. A3). In addition, seasonal minima in the NH occured one to two months later than observations, although the results in InvSURF were slightly better than in the prior and GOSAT inversions (Fig. A3). This indicates either possible errors in seasonal cycles of posterior emission, or atmospheric chemical sinks.

Comparison to ATom aircraft data also showed that on average, InvSURF, InvGLT and InvGLT_land resulted in the best statistics (Table 1), and the latitudinal gradient agreed better with the observations compared to InvGTOT below 4000 meter above sea level (masl) (Fig. 3b, A4 and A5). In this altitude zones, the mean bias was significantly larger in InvGTOT, which was in line with the results compared to surface ground-based stations (Table 1). This height corresponds approximately to the height where most of the surface ground-based stations were situated and from which JAXA/GOSAT XCH$_4$_LT data were calculated. Between 4000–8000 masl, the latitudinal gradients from InvGTOT were better captured whereas the other inversions overestimated these gradients (Fig. A4 and A5). Considering that the tropopause height is around 9000 masl or above, these results indicate that the transport model has problems in representing upper tropospheric concentrations. This is consistent with the finding that all model estimates worsened at high altitudes (> 8000 masl). All model estimates, both prior and posterior, failed to capture low mole fractions observed in high latitude (> 50° N/S) regions (Fig. A4). Such low CH$_4$ mole fractions were observed especially in the winter of 2017 and the spring of 2018 when the tropopause height was lower and the ATom aircraft operated in the stratosphere (Fig. A6). This is when the disagreement was strongest, possibly indicating the improper modelling of vertical profiles in polar vortex conditions.

### 3.3   Evaluation against TCCON

The latitudinal gradient was generally weaker in TCCON tropospheric partial and total columns (Fig. 3) compared to the surface observations, indicating a smaller influence of surface fluxes. In the northern high latitudes, the extracted TCCON partial columns successfully separated the stratospheric component, showing no significant low biases under polar vortex conditions (Fig. A7).

Compared to the surface data comparison, the TCCON comparison showed better overall agreement and improved representation of the latitudinal gradient in InvGTOT compared to other inversions for both total and tropospheric partial columns (Table 1, Fig. 3c and 3d). In the NH, all model estimates exhibited overestimation, driven by larger latitudinal gradients, with InvSURF, InvGLT and InvGLT_land performing worse than InvGTOT. Considering also that InvGTOT showed the best agreement with the comparison to Atom data between 4000–8000 masl on latitudinal gradients, this indicates the importance of the upper troposphere in the estimation of XCH$_4$. This finding also denotes that model biases in the estimation of XCH$_4$ are not solely caused by errors in resolving stratospheric concentrations.

### 3.4   CH$_4$ fluxes



**Table 2.** Global and regional $CH_4$ emissions (Tg $CH_4$ yr$^{-1}$), averaged over 2016–2019. Uncertainty presented in numbers after $\pm$ sign is the standard deviation of 500 ensemble members in CTE-$CH_4$.

| *Category* / Region | Prior | InvSURF | InvGLT | InvGLT_land | InvGTOT |
|---|---|---|---|---|---|
| *Total* | | | | | |
| Global | 523 ± 30 | 547 ± 26 | 550 ± 20 | 552 ± 21 | 544 ± 23 |
| 60° N–90° N | 17 ± 1 | 20 ± 1 | 18 ± 1 | 17 ± 1 | 17 ± 1 |
| 30° N–60° N | 179 ± 19 | 196 ± 14 | 191 ± 10 | 189 ± 10 | 154 ± 12 |
| EQ–30° N | 197 ± 18 | 200 ± 17 | 195 ± 13 | 199 ± 14 | 211 ± 15 |
| 30° S–EQ | 115 ± 13 | 115 ± 12 | 129 ± 10 | 130 ± 10 | 146 ± 11 |
| 90° S–30° S | 14 ± 2 | 15 ± 2 | 17 ± 2 | 17 ± 2 | 16 ± 2 |
| *Anthropogenic* | | | | | |
| Global | 356 ± 28 | 373 ± 67 | 382 ± 17 | 383 ± 18 | 368 ± 20 |
| 60° N–90° N | 5 ± 1 | 6 ± 1 | 5 ± 1 | 5 ± 1 | 5 ± 1 |
| 30° N–60° N | 150 ± 18 | 164 ± 14 | 161 ± 10 | 159 ± 10 | 125 ± 12 |
| EQ–30° N | 133 ± 16 | 136 ± 15 | 132 ± 11 | 135 ± 12 | 143 ± 13 |
| 30° S–EQ | 57 ± 11 | 57 ± 10 | 70 ± 8 | 70 ± 8 | 82 ± 9 |
| 90° S–30° S | 11 ± 2 | 12 ± 2 | 14 ± 2 | 14 ± 2 | 13 ± 2 |
| *Wetlands* | | | | | |
| Global | 120 ± 12 | 126 ± 11 | 121 ± 10 | 122 ± 10 | 130 ± 11 |
| 60° N–90° N | 8 ± 1 | 10 ± 1 | 8 ± 1 | 8 ± 1 | 8 ± 1 |
| 30° N–60° N | 21 ± 3 | 24 ± 2 | 22 ± 2 | 22 ± 2 | 22 ± 2 |
| EQ–30° N | 49 ± 7 | 49 ± 7 | 48 ± 6 | 49 ± 6 | 53 ± 7 |
| 30° S–EQ | 40 ± 6 | 40 ± 6 | 41 ± 6 | 41 ± 6 | 45 ± 6 |
| 90° S–30° S | 1 ± 1 | 1 ± 1 | 2 ± 1 | 2 ± 1 | 2 ± 1 |




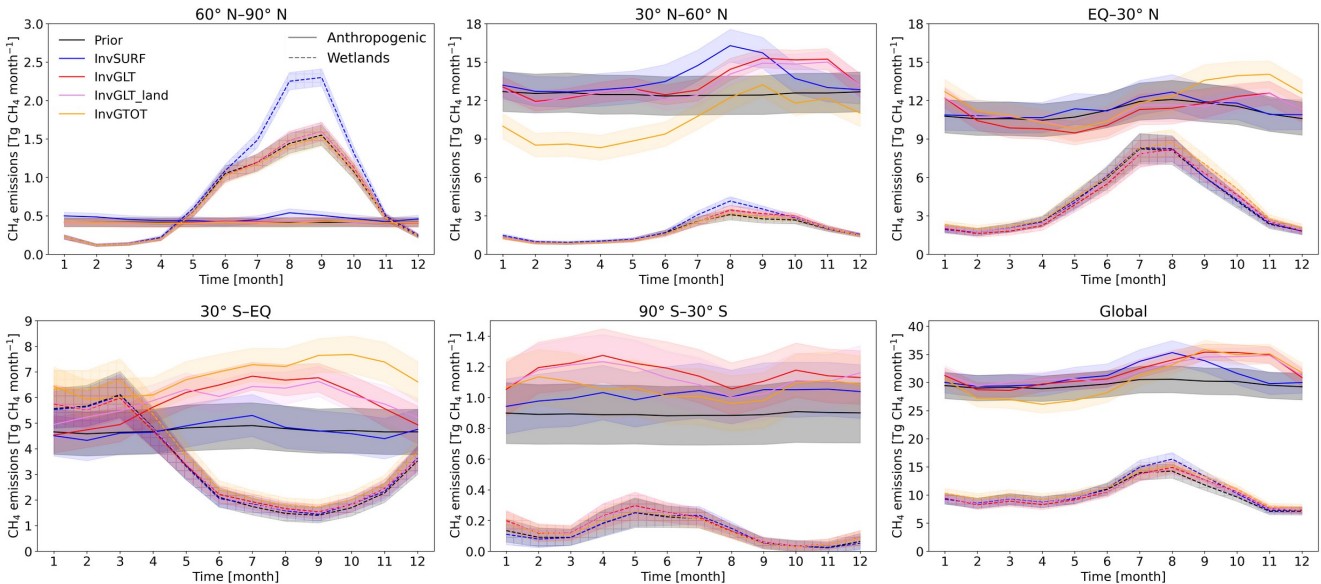

**Figure 4.** Monthly mean anthropogenic and wetland CH$_4$ emissions at 30° latitudinal bands and global totals. The means are taken from 2016–2019. Shaded areas illustrate uncertainty as standard deviations of 500 ensemble members.

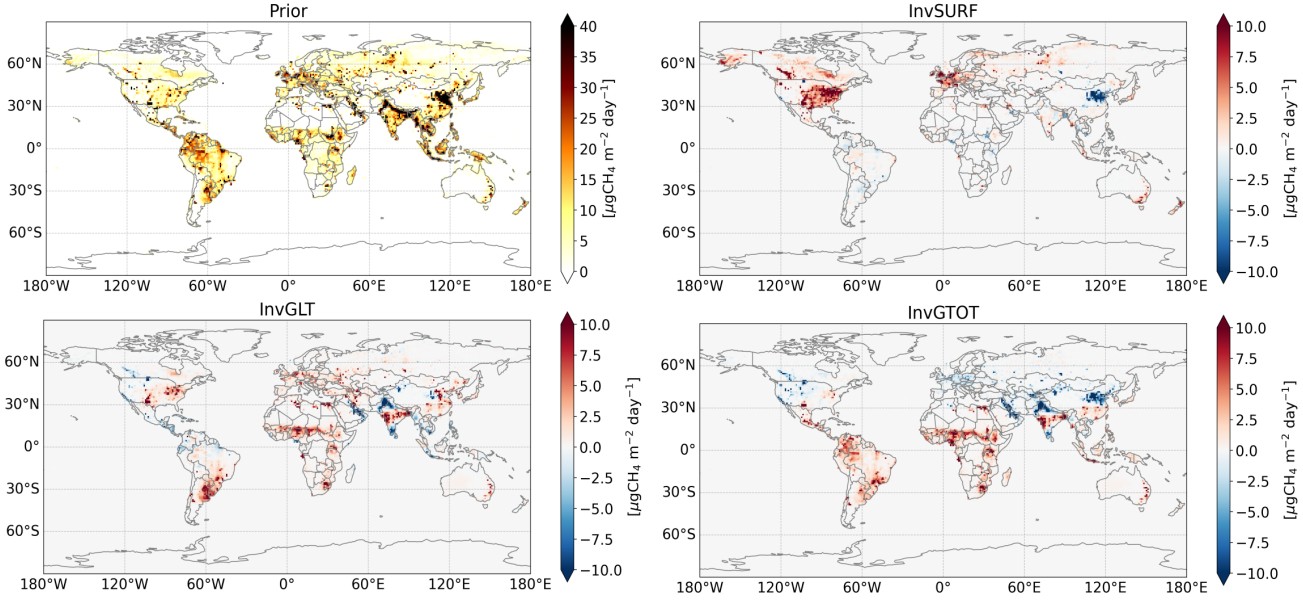

**Figure 5.** Prior total CH$_4$ emissions (left top) and emission increments (posterior - prior), averaged over 2016–2019. Positive values indicate posterior emissions being higher than the prior.





The 2016–2019 average global total and regional (30 latitudinal band) emissions are presented in Table 2. The posterior
global total emissions were similar in all inversions (544–552 $\pm$ 20–26 Tg CH$_4$ yr$^{-1}$) and showed increases from the prior
with reduction in uncertainty (523 $\pm$ 30 Tg CH$_4$ yr$^{-1}$). Most of the increase from the prior was attributed to anthropogenic
emissions (c.a. 57–96%). The sectoral emissions showed that anthropogenic emissions were higher and wetland emissions
lower in InvGLT and InvGLT_land compared to InvSURF and InvGTOT. The anthropogenic emissions in the SH in InvGLT
and InvGLT_land were higher compared to InvSURF but comparable in magnitude to InvSURF in the NH. InvGTOT also
showed higher anthropogenic emissions in the SH compared to InvSURF, especially in the tropics (30° S–EQ), but lower
anthropogenic emissions in 30° N–60° N.

Between the tropics and SH (90° S–30° N), the posterior anthropogenic emissions in the GOSAT inversions deviated more
from the prior compared to InvSURF (Table 2, Fig. 4). This difference was primarily associated with emissions over South
America, Africa and India (Fig. 5). The monthly variations of anthropogenic emissions deviated largely from the prior in the
GOSAT inversions (Fig. 4). All GOSAT inversion showed an emission peak in November in the region from the equator to
30° N, which was three months later than prior and InvSURF. In 30° S–EQ, all GOSAT inversions showed a clear seasonal
cycle with emissions peaking during April–November. In contrast, the prior had no clear seasonal cycle, and InvSURF showed
a small seasonal cycle with a peak in July. In 90° S–30° S, InvGLT and InvGLT_land showed two emission peaks, one around
April and another in October. InvGTOT did not show the April peak, but it aligned with InvGLT and InvGLT_land in presenting
an emission reduction in August and peak in October. InvSURF showed a less distinct seasonal cycle with emissions constantly
increasing from the prior in all months, except January. In the tropics, there are only a few surface stations (Fig. A1), which are
often far from emission sources, measuring background mixing air. As a result, the JAXA/GOSAT data likely contained more
valuable information to constrain the fluxes in these regions compared to the sparse surface data.

In 30° N–60° N, posterior anthropogenic emissions in InvGTOT were about 35 Tg CH$_4$ year$^{-1}$ lower than those in the
other inversions (Table 2), representing about 6% of the global total. These differences were not associated specifically with a
specific region, but InvGTOT showed mostly negative emission increments indicating that posterior emissions were lower than
the prior in this region (Fig. 5). Unlike the tropics, there are relatively large number of surface stations in Europe and the best
optimization setup in terms of model spatial resolution, and thus, we suspect that the results from InvGTOT in Europe may
have been too low. The comparison to ground-based observations in Europe also showed strong underestimation in InvGTOT
(-44 ppb on average compared to stations within [12° W–37° E, 34° N–73° N] domain). For other regions, considering that
the regions such as USA and China are large CH$_4$ emitters (Petrescu et al., 2024), we also suspect that the results in InvGTOT
were underestimated, possibly due to the ability of the transport model representation of XCH$_4$ (see also Section 4.1).

In 60° N–90° N, posterior wetland emissions in the GOSAT inversions stayed close to the prior compared to InvSURF (Table
2). While InvSURF showed an increase in summer wetland emissions, such change was not found in the GOSAT inversions
Therefore, the seasonal cycle amplitude of wetland emissions was smaller in the GOSAT inversions (Fig. 4). It is known that
the GOSAT data has sampling limitation during winter due to polar nights and very low solar zenith angles. However, JAXA/-
GOSAT data were available above 60° N during summer (Fig. 1). Further, as we found from the comparison of JAXA/GOSAT
retrievals to InvSURF (see Section 3.1), posterior mole fractions in InvSURF were higher than the JAXA/GOSAT retrievals,



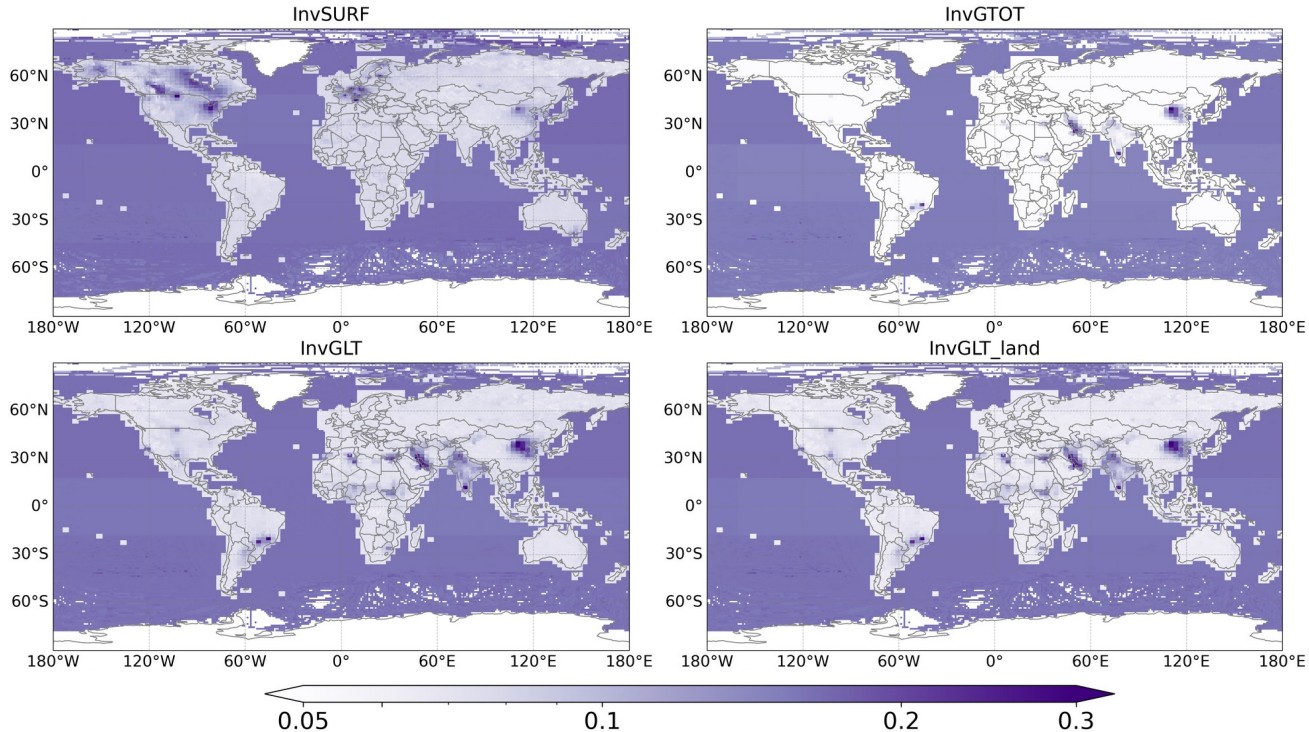

**Figure 6.** Annual mean uncertainty reduction rate $(\sigma_{\mathrm{prior}} - \sigma_{\mathrm{posterior}})/\sigma_{\mathrm{prior}}$ of total fluxes, averaged over 2016–2019. $\sigma$ were calculated as standard deviation from 500 ensemble members. Note the colour is in logarithmic scale.

indicating that emissions from the GOSAT inversions would likely be lower than those from InvSURF. Therefore, we argue that the lower emissions from the GOSAT inversions were not due to limited number of data, but rather reflected the agreement between the inversion and the prior.

## 3.5   Uncertainty estimates

Despite a rather arbitrary choice of prior observational uncertainty (diagonals of **R** in Eq. 1, see Section 2.1 and 2.3), the
posterior emission uncertainties were generally lower in the GOSAT inversions compared to InvSURF, and lowest in InvGLT. These differences were most pronounced in the NH tropics (EQ–30° N) (Table 2). The differences between InvSURF and GOSAT inversions were probably driven by the number of available observations, since GOSAT had much more data in the tropics. This could also explain in part the latitudinal bias found in Section 3.1. Within the GOSAT inversions, the number of assimilated data were higher in InvGTOT than InvGLT (Fig. A8), and the observational uncertainty and rejection thresh-
olds were the same in both inversions. This indicates that the lower uncertainty in InvGLT was not simply due to number of assimilated observations. However, it is possible that the prior observational uncertainty prescribed in InvGLT may have been underestimated, considering that retrieval errors are probably higher for lower tropospheric partial columns compare to




total columns. Consequently, the posterior uncertainty of the fluxes may have been also underestimated. On the other hand, the transport model error may be lower for the lower tropospheric partial column data, considering that the transport model performs better in representing tropospheric concentrations than stratospheric, so the total observational uncertainty could have been reasonable. The global total flux uncertainty of InvGLT_land was slightly higher compared to InvGLT. The posterior uncertainty was higher in the region with the largest uncertainty (EQ–30° N). Since the flux horizontal and seasonal distributions in InvGLT_land did not change significantly from InvGLT overall, we could argue that the effect of ocean data as a constraint was minor.

The spatial distribution of uncertainty reduction rates is shown in Fig. 6. This figure indicates that the GOSAT inversions constrained fluxes mostly in tropical regions and northeast China, while InvSURF showed strongest reductions in the USA, Canada and Europe. InvSURF also showed a reduction hot-spot areas in northeast China, but the signal was much weaker compared to the GOSAT inversions. The uncertainty reduction rates over land in InvGTOT was weak in general, with fewer reduction in hot-spots compared to InvGLT and InvGLT_land, indicating that the use of the lower tropospheric partial column data was more effective in constraining fluxes compared to total column data. The inversions with lower tropospheric partial column data showed stronger uncertainty reduction rates in Africa and India, which were not as prominent in other inversions. The weaker uncertainty reduction rates from the GOSAT inversions in North America and Europe were probably due to 1) larger uncertainty in the observations, where many surface observations in these regions had less than 30 ppb observation uncertainty (Table A1), and 2) lack of observations in high latitudes over winter.

## 4 Discussion

### 4.1 Role of upper atmosphere and OH in the estimation of $XCH_4$ values

Latitudinal differences between posterior from the surface inversion and satellite total column retrievals have been seen in earlier studies using TM5 and other global Eulerian atmospheric chemistry transport models (Alexe et al., 2015; Qu et al., 2021; Tsuruta et al., 2017; Turner et al., 2015; Zhang et al., 2021). Such discrepancies were reported regardless of the satellites' retrieval products, prior fluxes, years or seasons. Part of the misrepresentation of $CH_4$ mole fractions in the upper troposphere could be due to convection schemes in the transport models (Saito et al., 2013), but the exact effect on the representation of total column values are to be examined. In high northern latitudes, the stratospheric profile is one of the challenges especially in polar vortex conditions (Tsuruta et al., 2023). However, polar vortex occurs only occasionally in spring, and our comparison to GOSAT, TROPOMI and TCCON $XCH_4$ values in this and previous studies showed biases even during the periods without polar vortex conditions (Tsuruta et al., 2017, 2023). As shown in this study, latitudinal biases occur already in the upper troposphere (Fig. A5), which confirms the findings from Lindqvist et al. (2024) who argued that the role of the stratosphere in the estimation of $XCH_4$ was minor.

In addition, we found that using higher horizontal resolution in TM5 improves agreement in the lower tropospheric partial and total columns (Fig. A2), indicating the importance of the transport model resolution. The latitudinal biases indeed seem to be slightly lower using higher spatial resolution (Stanevich et al., 2021), although exact effect is to be examined. This study was





limited in its exploration of the impact of the horizontal resolution, but increasing model resolution in the vertical dimension could also improve the representation of upper atmospheric $CH_4$.

The discrepancies in the latitudinal gradient could also be caused by the choice of chemistry schemes in the transport models. The distribution of OH, the largest sink of $CH_4$ (Saunois et al., 2024), plays an important role in regulating ambient methane

levels. The OH concentration fields by Spivakovsky et al. (2000) used in this study and several others (Patra et al., 2011; Saunois et al., 2024) have relatively uniform inter-hemispheric distributions, possibly underestimating the OH concentration in the NH and overestimating them in the SH (Zhao et al., 2019). This leads to higher atmospheric $CH_4$ in the NH and lower in the SH, as shown by this and previous studies (Tsuruta et al., 2017, 2023). Zhao et al. (2020) showed that the differences in the inter-hemispheric distribution of OH could lead to about 25–50 Tg $CH_4$ yr$^{-1}$ differences in the inter-hemispheric distribution of

$CH_4$ emissions, where Spivakovsky et al. (2000) based inversion lead to lower emissions in the NH and higher in the SH. This outcome is in line with the conclusion that the emission distributions in InvGTOT, which estimated the lowest $CH_4$ emissions in the NH, could be unreliable. The vertical OH profiles used in this study that have distinct peaks at around 500–600 hPa (Zhao et al., 2019) may also partially explain why ATom profiles and GOSAT total columns were not accurately reproduced.

### 4.2   Land-sea discrepancies

In $CH_4$ inversions using the GOSAT data, it has not been common to correct land-sea biases in the retrievals or exclude data over ocean. $CH_4$ fluxes over the ocean are minor compared to those over land (Saunois et al., 2024). Consequently, the effect of land-seas biases in the JAXA/GOSAT retrieval data is expected to be small in estimation of $CH_4$ fluxes. This study showed that the emission estimates and posterior mole fractions from InvGLT and InvGLT_land were very similar, despite the different systematic and seasonal biases over land and sea compared to InvSURF (Fig. 2), confirming that the effect of ocean data as

constraints has minimum influence to the outcome of inversions. Therefore, to significantly decrease the amount of data in the inversions and increase the computational efficiency.

### 4.3   Global and regional emissions

The global total estimates from InvSURF were slightly lower than from the inversions using JAXA/GOSAT_LT data, and on the lower edge of the range of the top-down (TD) estimates from the latest Global Methane Budgets (553–586 Tg $CH_4$ yr$^{-1}$,

2010–2019 average; Saunois et al., 2024). The breakdown of anthropogenic and wetlands sources showed that anthropogenic emissions from this study were slightly higher compared to Saunois et al. (2024) (350–391 Tg $CH_4$ yr$^{-1}$ vs 145–214 Tg $CH_4$ yr$^{-1}$, 2010–2019 TD averages). These differences is probably because of smaller wetland prior emissions (120 Tg $CH_4$ yr$^{-1}$), which are on the lower boundary of bottom-up estimates from Saunois et al. (2024) (119–203 Tg $CH_4$ yr$^{-1}$, 2010–2019 bottom-up averages). This indicates that the prior uncertainty may need to be revised, and for example, spatially and seasonally

varying uncertainty ratios Tenkanen et al. (2024) could provide better freedom in the inversion. It is worth pointing out that we did not use a separate prior for freshwater emissions, and, thus, the freshwater emissions could be wrongly attributed to anthropogenic emissions, if there is a spatial overlap with the anthropogenic emissions.



The total emissions in the eastern part of North America agree well with previous studies (Alexe et al., 2015; Baray et al., 2021; Lu et al., 2022). Higher emissions in the northeastern part of the US compared to EDGAR (Fig. 5) point to underesti-
395 mation of emissions from oil and gas. It should be noted that although we used the newer version of EDGAR (v8.0) in this study, the underestimation still seems to remain. The seasonal variability of anthropogenic $CH_4$ emissions in southern North America (including the contiguous United States and Mexico) from the GOSAT inversions (Fig. A9a) showed opposite patterns compared to InvSURF. The seasonal cycle in this region is likely to be associated with natural gas consumption (Zeng et al., 2023) and agriculture (Maasakkers et al., 2023). In contrast to results from Maasakkers et al. (2023) who argue that the
400 seasonal pattern of natural gas consumption has strong interannual variations and is spatially inhomogeneous, our results from the GOSAT inversions were similar to those from Miller et al. (2013) which showed larger emissions during autumn and winter compared to summer.

In China, InvSURF and InvGTOT showed strong negative emission increments around Beijing, which is consistent with studies such as Lu et al. (2021) and Wang et al. (2022) who assimilated both surface and the GOSAT $XCH_4$ data. This seems
to be a common feature found in other studies despite different GOSAT retrieval products, prior, transport and inverse models were used (Lu et al., 2021, and references therein). On the other hand, InvGLT and InvGLT_land showed positive emission increments in eastern and southern China. This is closer to the findings from Chen et al. (2022) who assimilated the TROPOMI $XCH_4$ data. However, these results again contradict with Qu et al. (2021) who assimilated both GOSAT and TROPOMI $XCH_4$ data and argued that an inversion using only TROPOMI $XCH_4$ may be unreliable over China because the TROPOMI $XCH_4$ is
strongly affected by cloud cover.

In central Africa and South Sudan wetland regions, our GOSAT inversions showed a large increase in total emissions (Fig. 5), which were primarily associated with wetland emissions. This is in line with findings from Lunt et al. (2019), who used the GOSAT $XCH_4$, and Pandey et al. (2021), who used the TROPOMI $XCH_4$ in their inversions, although our posterior estimates were considerably lower. For instance, we found monthly maxima below 1 Tg $CH_4$ month$^{-1}$ around South Sudan, compared to
415 Lunt et al. (2019) estimates of 7 Tg $CH_4$ month$^{-1}$ and Pandey et al. (2021) estimates of seasonal cycle amplitude of about 2–3 Tg $CH_4$ month$^{-1}$. Nevertheless, the seasonal cycle in this region from the GOSAT inversions (Fig. A9b) showed significantly lower emissions in June–July and later maxima in August–September compared to prior and InvSURF. This corresponds slightly better to dry and wet seasons and is in line with Lunt et al. (2019) and Pandey et al. (2021), although we could not reproduce high emissions in late months.

The magnitude of high northern latitude (NHL) wetlands remained uncertain. Previous studies showed that in some cases, NHL emissions from the inversion based on the GOSAT data were lower compared to those based on surface data (Pandey et al., 2016; Feng et al., 2017), while others showed the opposite (Alexe et al., 2015; Baray et al., 2021). Our analysis showed that the wetland emissions for 60° N–90° N were lower in the GOSAT inversions regardless of the assimilated data type (total or lower tropospheric partial column) compared to InvSURF. This is in line with our previous studies assimilating TROPOMI
$XCH_4$ data (Tsuruta et al., 2023), and using other wetland priors, such as JSBACH-HIMMELI, where emissions were relatively larger among process-based models (Aalto et al., 2024; Tenkanen et al., 2024). Tsuruta et al. (2023) assumed that the lower NHL wetland emissions in the TROPOMI inversions were due to latitudinal biases associated with model-retrieval differences



(Lindqvist et al., 2024). However, as shown in this study, the inversion using lower tropospheric partial column data also resulted in lower NHL wetland emissions compared to InvSURF, indicating that there may be fundamental discrepancies between satellite and surface inversions. The differences were partly due to spatial coverages and temporal distributions of the observations and uncertainties associated with transport models, observations, prior emissions and possible biases in the satellite data, but further study is needed to find the exact cause of these discrepancies and to obtain more robust the emission estimates.

## 5 Conclusions

This study presented the advantages of JAXA/GOSAT lower tropospheric partial column retrievals in estimating global and regional $CH_4$ budgets using the CTE-$CH_4$ atmospheric inverse model. Our findings showed that assimilating the lower tropospheric partial column data led to posterior $CH_4$ fluxes and atmospheric $CH_4$ mole fractions that were more consistent with the inversion estimates using surface data compared to total column retrievals. In addition, partial column retrievals better constrained $CH_4$ fluxes in low latitudes than surface data with sparse observation network. This is a considerable advantage to the atmospheric inverse modelling community, and the partial column product could potentially be a better product than total column for estimation of $CH_4$ fluxes.

In addition, we found that lower tropospheric partial column data possibly reduce global emission uncertainty. In addition, it was concluded that the lower tropospheric partial column ocean data have minimal influence on constraining $CH_4$ fluxes over land, suggesting that excluding ocean data could improve computational efficiency. However, further studies are needed to assess uncertainty in the partial column retrievals and transport model's ability to represent partial column mole fractions. This study also highlighted the importance of transport model resolution in estimation of total and partial column data, indicating the need for high resolution transport models in satellite-driven inversions.

Our study was limited to retrievals from one satellite (GOSAT) assimilated to a single inverse model (CTE-$CH_4$) for a relatively short period (four years). Future efforts should focus on exploiting the use of other satellite datasets and inverse models. JAXA/GOSAT provides partial column products since 2009 up to today, including those from GOSAT-2 since 2019, that will allow expanding the study period to perform trend analysis. In addition, in this study, we did not take into account averaging kernels and detailed retrieval uncertainty, which is available in the newest version of JAXA/GOSAT partial column product (v3.0). Including this information will possibly result in more realistic estimates of $CH_4$ fluxes and their uncertainties. Lastly, JAXA/GOSAT provides GOSAT partial column products for upper layers (upper tropospheric and three stratospheric layers). The use of these products in inversions can provide useful information in identifying the causes of transport model biases in vertical layers.

*Code and data availability.* All the model results, inputs and code will be provided on request from the corresponding author (Aki Tsuruta, aki.tsuruta@fmi.fi).



*Author contributions.* The research was conceptualized by HS with contributions from AT and TA. AT developed model codes for assim-
ilation of lower tropospheric partial column data, and carried out all inverse modelling simulations and visualization. AT did analysis with
contributions from AK, KSh, FK, TA, MKT, LF, EK and HS. AK, KSh, FK, NK and HS provided the GOSAT data. EK calculated modelled
$XCH_4$ values at TCCON sites. LF calculated TCCON tropospheric partial columns. KM provided ATom data. TCCON data were provided by
OEG (Izaña), FH (Karlsruhe), RK (Sodankylä), IM (Tsukuba), HO (Rikubetsu), DFP (Lauder), KSh (Saga), KSt (Eureka), RS (Garmisch),
MKS (Reunion Island), YT (Paris), VAV (Burgos), MV (Nicosia), TW (Orleans) and MZ (Xianghe). AT wrote the original draft of the
manuscript with contributions from AK, LB, EK, MKT and HS. All authors have read and approved to publish the version of the manuscript.

*Competing interests.* The contact author has declared that none of the authors has any competing interests.

*Financial support.* We thank the JAXA (24RT000409), EU-H2020 VERIFY (776810), CoCO$_2$ (958927) and EMME-CARE – Eastern
Mediterranean and Middle East – Climate and Atmosphere Research Centre (856612), EU-Horizon EYE-CLIMA (101081395) and IM4CA
(101183460), Finnish Center of Excellences (272041, 353082), Research Council of Finland projects FIRI - ICOS Finland (345531), GHG-
SUPER (351311) and CHARM (364975), Research Council of Finland Flagships ACCC (337552) and FAME (359196), CSC (FICOCOSS)
and European Space Agency projects AMPAC-Net (AO/1-10901/21/I-D) and SMART-CH4 (AO/1-11844/23/I-NS) for financial support.
Measurements at Jungfraujoch were supported by ICOS Switzerland (SNF grant 20F120_198227). The Paris site has received funding from
Sorbonne Université, the French research center CNRS and the French space agency CNES. The Eureka TCCON measurements were made
at the Polar Environment Atmospheric Research Laboratory (PEARL), primarily supported NSERC, ECCC, and CSA. The TCCON sites at
Rikubetsu, Tsukuba, and Burgos are supported in part by the GOSAT series project. Local support for Burgos is provided by the Energy De-
velopment Corporation (EDC, Philippines). The Réunion Island station is operated by the Royal Belgian Institute for Space Aeronomy with
financial support since 2014 by the EU project ICOS-Inwire (Grant agreement ID 313169), the ministerial decree for ICOS (FR/35/IC1 to
FR/35/IC6), ESFRI-FED ICOS-BE (EF/211/ICOS-BE) project and local activities supported by LACy/UMR8105 and by OSU-R/UMS3365
– Université de La Réunion.

*Acknowledgements.* We thank the NOAA ObsPack and ICOS PIs for providing the valuable global CH$_4$ mole fractions data. We are grateful
for the NOAA Global Monitoring Laboratory (NOAA/GML), CSIRO Oceans and Atmosphere, Climate Science Centre (CSIRO), Umwelt-
bundesamt Austria/Environment Agency Austria (EAA), Environment and Climate Change Canada (ECCC), the Finnish Meteorological
Institute (FMI), Commissariat à l'énergie atomique et aux énergies alternatives (CEA), Atmospheric Sciences and Climate (ISAC), Ente
per le Nuove Tecnologie, l'Energia e l'Ambiente (ENEA), International Foundation High Altitude Research Stations Jungfraujoch and
Gornergrat (HFSJG), the Institute for the Institute on Atmospheric Pollution of the National Research Council (IIA), the Institute of Atmo-
spheric Sciences and Climate (ISAC), the Environment Division Global Environment and Marine Department Japan Meteorological Agency
(JMA), Joint Research Centre (JRC), Lawrence Berkeley National Laboratory (LBNL-ARM), Laboratoire des Sciences du Climat et de
l'Environnement (LSCE), University of Lund (LUND-CEC), the Max Planck Institute for Biogeochemistry (MPIBGC), National Institute
for Environmental Studies (NIES), Norwegian Institute for Air Research (NILU), the Pennsylvania State University (PSU), Swedish Univer-



sity of Agricultural Sciences (SLU), University of Bristol (UNIVBRIS), University of Eastern Finland (UEF), University of Exeter (Univ. Exeter), University of Groningen (RUG) and University of Helsinki (UHELS) for performing high-quality $CH_4$ measurements at global sites and making them available through the NOAA ObsPack and personal communications. The authors would like to thank the ICOS PIs for providing atmospheric $CH_4$ data (see Table A2). We thank TCCON PIs for providing validation data: Justus Notholt, Institute of Environmental Physics, University of Bremen, Bremen, Germany (Bremen site), Paul O. Wennberg, Division of Engineering and Applied

Science, California Institute of Technology, Pasadena, CA, USA (California Institute of Technology, Lamont, Park Falls sites), Nicholas M. Deutscher, Centre for Atmospheric Chemistry, School of Earth, Atmospheric and Life Sciences, University of Wollongong, Wollongong, Australia (Darwin and Wollongong sites), Laura T. Iraci, NASA Ames Research Center, Moffett Field, CA, USA (Armstrong Flight Research Center and Indianapolis sites), Debra Wunch, Department of Physics, University of Toronto, Toronto, Ontario, Canada (East Trout Lake site), Wei Wang, Key Laboratory of Environmental Optics and Technology, Anhui Institute of Optics and Fine Mechanics, Hefei, China (Hefei

site) and Martin Steinbacher, Empa, Swiss Federal Laboratories for Materials Science and Technology (Ny-Ålesund site).

**Appendix A:  Appendix A**



**Table A1.** The ground-based in-situ measurement sites used in InvSURF and evaluation. Observation uncertainty (Obs. Unc.) is sum of measurement and transport model errors used in the observation covariance matrix. Data type is categorized into two: discrete (D) and continuous (C).

| ID | Sites | Country/Territory | Laboratory | Longitude [deg. E] | Latitude [deg. N] | Altitude [m a.s.l.] | Obs. Unc. [ppb] | Data type |
|---|---|---|---|---|---|---|---|---|
| ABT | Abbotsford, British Columbia | Canada | ECCC | -122.34 | 49.01 | 93 | 30 | C |
| ALT | Alert, Nunavut | Canada | NOAA | -62.51 | 82.45 | 190 | 15 | D |
| ALT | Alert, Nunavut | Canada | ECCC | -62.51 | 82.45 | 195 | 15 | C |
| AMY | Anmyeon-do | Republic of Korea | NOAA | 126.33 | 36.54 | 87 | 30 | D |
| ASC | Ascension Island | United Kingdom | NOAA | -14.4 | -7.97 | 90 | 15 | D |
| ASK | Assekrem | Algeria | NOAA | 5.63 | 23.26 | 2715 | 25 | D |
| AZR | Terceira Island, Azores | Portugal | NOAA | -27.36 | 38.76 | 24 | 15 | D |
| AZV | Azovo | Russian Federation | NIES | 73.03 | 54.71 | 190 | 30 | C |
| BAR | Baranova | Russian Federation | FMI | 101.62 | 79.28 | 30 | 4.5 | C |
| BCK | Behchoko, Northwest Territories | Canada | ECCC | -115.92 | 62.8 | 220 | 15 | C |
| BHD | Baring Head Station | New Zealand | NOAA | 174.87 | -41.41 | 90 | 4.5 | D |
| BKT | Bukit Kototabang | Indonesia | NOAA | 100.31 | -0.2 | 875 | 75 | D |
| BLK | Baker Lake, Nunavut | Canada | ECCC | -96.01 | 64.33 | 61 | 15 | C |
| BMW | Tudor Hill, Bermuda | United Kingdom | NOAA | -64.88 | 32.26 | 33 | 15 | D |
| BRA | Bratt's Lake Saskatchewan | Canada | ECCC | -104.71 | 50.2 | 630 | 75 | C |
| BRW | Barrow Atmospheric Baseline Observatory | United States | NOAA | -156.61 | 71.32 | 27 | 15 | C |
| BRW | Barrow Atmospheric Baseline Observatory | United States | NOAA | -156.58 | 71.32 | 16 | 15 | D |
| BRZ | Berezorechka | Russian Federation | NIES | 84.33 | 56.15 | 248 | 75 | C |
| BSD | Bilsdale | United Kingdom | UNIVBRIS | -1.15 | 54.36 | 628 | 30 | C |
| CBA | Cold Bay, Alaska | United States | NOAA | -162.71 | 55.21 | 25 | 15 | D |
| CBY | Cambridge Bay, Nunavut Territory | Canada | ECCC | -105.06 | 69.13 | 47 | 15 | C |
| CFA | Cape Ferguson | Australia | CSIRO | 147.06 | -19.28 | 5 | 25 | D |
| CGO | Cape Grim, Tasmania | Australia | NOAA | 144.68 | -40.68 | 164 | 4.5 | D |
| CGO | Cape Grim | Australia | CSIRO | 144.68 | -40.68 | 94 | 15 | C |
| CGR | Charles Point, Darwin | Australia | CSIRO | 12.65 | 37.67 | 9 | 25 | C |
| CHL | Churchill, Manitoba | Canada | ECCC | -93.82 | 58.74 | 89 | 15 | C |
| CHR | Christmas Island | Republic of Kiribati | NOAA | -157.15 | 1.7 | 5 | 15 | D |
| CMN | Mt. Cimone Station | Italy | ICOS-ATC,CNR-ISAC | 10.7 | 44.19 | 2173 | 15 | C |
| CPS | Chapais,Quebec | Canada | ECCC | -74.98 | 49.82 | 431 | 15 | C |
| CPT | Cape Point | South Africa | NOAA | 18.49 | -34.35 | 260 | 25 | D |
| CRV | Carbon in Arctic Reservoirs Vulnerability Expe... | United States | NOAA | -147.6 | 64.99 | 643 | 15 | C |
| CRZ | Crozet Island | France | NOAA | 51.85 | -46.43 | 202 | 4.5 | D |
| CUR | Monte Curcio | Italy | IIA | 16.42 | 39.32 | 1801 | 15 | C |
| CYA | Casey Station, Antarctica | Australia | CSIRO | 110.52 | -66.28 | 55 | 4.5 | D |
| DEM | Demyanskoe | Russian Federation | NIES | 70.87 | 59.79 | 155 | 30 | C |
| DRP | Drake Passage | Drake Passage | NOAA | -61.68 | -59.07 | 10 | 4.5 | D |
| DSI | Dongsha Island | Taiwan | NOAA | 116.73 | 20.7 | 8 | 15 | D |
| DVV | Danville, Virginia | United States | PSU | -79.44 | 36.71 | 492 | 15 | C |
| EGB | Egbert, Ontario | Canada | ECCC | -79.78 | 44.23 | 276 | 25 | C |
| EIC | Easter Island | Chile | NOAA | -109.45 | -27.13 | 72 | 4.5 | D |
| ENA | Eastern North Atlantic, Graciosa, Azores | Portugal | LBNL-ARM | -28.03 | 39.09 | 40 | 25 | C |





**Table A1.** Continuation to Table A1.

| | | | | | | | | |
|---|---|---|---|---|---|---|---|---|
| ESP | Estevan Point, British Columbia | Canada | ECCC | -126.54 | 49.38 | 47 | 25 | C |
| EST | Esther, Alberta | Canada | ECCC | -110.21 | 51.67 | 757 | 30 | C |
| ETL | East Trout Lake, Saskatchewan | Canada | ECCC | -104.99 | 54.35 | 598 | 30 | C |
| FNE | Fort Nelson, British Columbia | Canada | ECCC | -122.57 | 58.84 | 376 | 30 | C |
| FSD | Fraserdale | Canada | ECCC | -81.57 | 49.88 | 250 | 30 | C |
| GAT | Gartow | Germany | ICOS-ATC,HPB | 11.44 | 53.07 | 411 | 25 | C |
| GCI | Millerville, AL | United States | PSU | -85.89 | 33.18 | 428 | 25 | C |
| GMI | Mariana Islands | Guam | NOAA | 144.66 | 13.39 | 8 | 15 | D |
| GPA | Gunn Point | Australia | CSIRO | 131.04 | -12.25 | 37 | 75 | D |
| HBA | Halley Station, Antarctica | United Kingdom | NOAA | -26.21 | -75.61 | 35 | 4.5 | D |
| HNP | Hanlan's Point, Ontario | Canada | ECCC | -79.39 | 43.61 | 97 | 25 | C |
| HPB | Hohenpeissenberg | Germany | ICOS-ATC,HPB | 11.02 | 47.8 | 1065 | 25 | C |
| HSU | Humboldt State University | United States | NOAA | -124.44 | 41.57 | 8 | 30 | D |
| HTM | Hyltemossa | Sweden | ICOS-ATC,LUND-CEC | 13.42 | 56.1 | 265 | 25 | C |
| ICE | Storhofdi, Vestmannaeyjar | Iceland | NOAA | -20.29 | 63.4 | 122 | 15 | D |
| INU | Inuvik,Northwest Territories | Canada | ECCC | -133.53 | 68.32 | 123 | 15 | C |
| IPR | Ispra | Italy | ICOS-ATC,JRC | 8.64 | 45.81 | 310 | 30 | C |
| IZO | Izana, Tenerife, Canary Islands | Spain | NOAA | -16.48 | 28.3 | 2378 | 25 | D |
| JFJ | Jungfraujoch | Switzerland | ICOS-ATC,HFSJG | 7.99 | 46.55 | 3585 | 15 | C |
| KEY | Key Biscayne, Florida | United States | NOAA | -80.2 | 25.67 | 6 | 25 | D |
| KIT | Karlsruhe | Germany | ICOS-ATC,HPB | 8.42 | 49.09 | 310 | 30 | C |
| KJN | Kjölnes | Norway | Univ. Exeter | 29.23 | 70.85 | 20 | 15 | C |
| KMP | Kumpula | Finland | FMI | 24.96 | 60.2 | 53 | 30 | C |
| KRE | Kresin u Pacova | Czech Republic | ICOS | 15.08 | 49.57 | 784 | 25 | C |
| KRS | Karasevoe | Russian Federation | NIES | 82.42 | 58.25 | 156 | 30 | C |
| KUM | Cape Kumukahi, Hawaii | United States | NOAA | -155.01 | 19.51 | 3 | 15 | D |
| LEF | Park Falls, Wisconsin | United States | NOAA | -90.27 | 45.95 | 868 | 30 | C |
| LIN | Lindenberg | Germany | ICOS-ATC,HPB | 14.12 | 52.17 | 171 | 30 | C |
| LLB | Lac La Biche, Alberta | Canada | ECCC | -112.47 | 54.95 | 590 | 30 | C |
| LLN | Lulin | Taiwan | NOAA | 120.86 | 23.47 | 2867 | 25 | D |
| LMP | Lampedusa | Italy | ICOS-ATC,ENEA | 12.63 | 35.52 | 53 | 25 | C |
| LMT | Lamezia Terme | Italy | ISAC | 16.23 | 38.88 | 14 | 30 | C |
| LUT | Lutjewad | Netherlands | ICOS-ATC,RUG | 6.35 | 53.4 | 61 | 25 | C |
| MAA | Mawson, Antarctica | Australia | CSIRO | 62.87 | -67.62 | 32 | 4.5 | D |
| MEX | High Altitude Global Climate Observation Center | Mexico | NOAA | -97.31 | 18.98 | 4469 | 15 | D |
| MID | Sand Island, Midway | United States | NOAA | -177.38 | 28.21 | 8 | 15 | D |
| MLO | Mauna Loa, Hawaii | United States | NOAA | -155.58 | 19.54 | 3437 | 15 | C/D |
| MNM | Minamitorishima | Japan | JMA | 153.98 | 24.29 | 27 | 15 | C |
| MQA | Macquarie Island | Australia | CSIRO | 158.97 | -54.48 | 13 | 4.5 | D |
| MRC | Marcellus Pennsylvania | United States | PSU | -76.42 | 41.47 | 652 | 75 | C |



**Table A1.** Continuation of Table A1.

| NAT | Farol De Mae Luiza Lighthouse | Brazil | NOAA | -35.19 | -5.51 | 20 | 15 | D |
|-----|-------------------------------|--------|------|--------|-------|----|----|---|
| NMB | Gobabeb | Namibia | NOAA | 15.01 | -23.58 | 461 | 25 | D |
| NOR | Norunda | Sweden | ICOS-ATC,LUND-CEC | 17.48 | 60.09 | 146 | 15 | C |
| NOY | Noyabrsk | Russian Federation | NIES | 75.78 | 63.43 | 188 | 30 | C |
| NWR | Niwot Ridge, Colorado | United States | NOAA | -105.57 | 40.05 | 3526 | 15 | D |
| OPE | Observatoire perenne de l'environnement | France | ICOS-ATC,LSCE | 5.5 | 48.56 | 510 | 30 | C |
| OXK | Ochsenkopf | Germany | ICOS-ATC,CAL-FCL | 11.81 | 50.03 | 1185 | 30 | C |
| PAL | Pallas-Sammaltunturi, GAW Station | Finland | ICOS-ATC,FMI | 24.12 | 67.97 | 577 | 15 | C |
| PDM | Pic du Midi | France | LSCE | 0.14 | 42.94 | 2887 | 15 | D |
| POC | Pacific Ocean | Pacific Ocean | NOAA | -130.75 | 0.12 | 20 | 15 | D |
| PSA | Palmer Station, Antarctica | United States | NOAA | -64.05 | -64.77 | 15 | 4.5 | D |
| PUI | Puijo | Finland | ICOS-ATC,UEF | 27.66 | 62.91 | 84 | 30 | C |
| PUY | Puy de Dome | France | ICOS-ATC,LSCE | 2.97 | 45.77 | 1475 | 15 | C |
| RPB | Ragged Point | Barbados | NOAA | -59.43 | 13.16 | 20 | 15 | D |
| RUN | La Réunion | France | ICOS-ATC,LSCE | 55.38 | -21.08 | 2160 | 15 | C |
| RYO | Ryori | Japan | JMA | 141.82 | 39.03 | 280 | 15 | C |
| SAC | Saclay | France | ICOS-ATC,CEA | 2.14 | 48.72 | 260 | 75 | C |
| SCT | Beech Island, South Carolina | United States | NOAA | -81.83 | 33.41 | 420 | 75 | C |
| SDZ | Shangdianzi | China | NOAA | 117.12 | 40.65 | 298 | 15 | D |
| SEY | Mahe Island | Seychelles | NOAA | 55.53 | -4.68 | 7 | 15 | D |
| SGP | Southern Great Plains, Oklahoma | United States | NOAA | -97.5 | 36.62 | 339 | 75 | D |
| SGP | Southern Great Plains, Oklahoma | United States | LBNL-ARM | -97.49 | 36.61 | 374 | 75 | C |
| SHM | Shemya Island, Alaska | United States | NOAA | 174.08 | 52.72 | 28 | 25 | D |
| SMO | Tutuila | American Samoa | NOAA | -170.56 | -14.23 | 60 | 15 | D |
| SMR | Hyytiala | Finland | ICOS-ATC,UHELS | 24.29 | 61.85 | 306 | 25 | C |
| SNB | Sonnblick | Austria | EAA | 47.05 | 12.96 | 3111 | 15 | C |
| SOD | Sodanyklä | Finland | FMI | 26.64 | 67.36 | 227 | 25 | C |
| SPO | South Pole, Antarctica | United States | NOAA | -24.8 | -89.96 | 2821 | 4.5 | D |
| STE | Steinkimmen | Germany | ICOS-ATC,HPB | 8.46 | 53.04 | 281 | 75 | C |
| SUM | Summit | Greenland | NOAA | -38.42 | 72.6 | 3215 | 15 | D |
| SVB | Svartberget | Sweden | ICOS-ATC,SLU | 19.77 | 64.26 | 419 | 25 | C |
| SYO | Syowa Station, Antarctica | Japan | NOAA | 39.59 | -69 | 16 | 4.5 | D |
| TAC | Tacolneston | United Kingdom | NOAA | 1.14 | 52.52 | 236 | 25 | D |
| TAP | Tae-ahn Peninsula | Republic of Korea | NOAA | 126.13 | 36.73 | 21 | 75 | D |
| THD | Trinidad Head, California | United States | NOAA | -124.15 | 41.05 | 112 | 25 | D |
| TIK | Hydrometeorological Observatory of Tiksi | Russia | NOAA | 128.89 | 71.6 | 29 | 15 | D |
| TIK | Tiksi | Russian Federation | FMI | 128.89 | 71.6 | 29 | 15 | C |





**Table A1.** Continuation of Table A1.

| | | | | | | | | |
|---|---|---|---|---|---|---|---|---|
| TOH | Torfhaus | Germany | ICOS-ATC,HPB | 10.53 | 51.81 | 948 | 25 | C |
| TPD | Turkey Point, Ontario | Canada | ECCC | -80.56 | 42.64 | 266 | 25 | C |
| TRN | Trainou | France | ICOS-ATC,LSCE | 2.11 | 47.96 | 311 | 25 | C |
| USH | Ushuaia | Argentina | NOAA | -68.31 | -54.85 | 32 | 4.5 | D |
| UTA | Wendover, Utah | United States | NOAA | -113.72 | 39.9 | 1332 | 25 | D |
| UTO | Uto | Finland | ICOS-ATC,FMI | 21.37 | 59.78 | 65 | 25 | C |
| UUM | Ulaan Uul | Mongolia | NOAA | 111.1 | 44.45 | 1012 | 25 | D |
| VGN | Vaganovo | Russian Federation | NIES | 62.32 | 54.5 | 277 | 30 | C |
| WIS | Weizmann Institute of Science at the Arava Ins... | Israel | NOAA | 35.06 | 29.96 | 482 | 25 | D |
| WLG | Mt. Waliguan | Peoples Republic of China | NOAA | 100.9 | 36.27 | 3890 | 15 | D |
| WSA | Sable Island, Nova Scotia | Canada | ECCC | -60.01 | 43.93 | 8 | 25 | C |
| YON | Yonagunijima | Japan | JMA | 123.01 | 24.47 | 50 | 30 | C |
| ZEP | Ny-Alesund, Svalbard | Norway and Sweden | ICOS-ATC,NILU | 11.89 | 78.91 | 489 | 15 | C |
| ZOT | Zotino | Russian Federation | MPIBGC | 89.21 | 60.48 | 415 | 25 | C |



**Table A2.** List of the ICOS sites, data categories, PIs and references for the sites used in InvSURF and evaluation.

| Site code | Data category | Authors | References |
|---|---|---|---|
| CMN | ICOS ATC NRT CH4 growing time series | Cristofanelli, Paolo and Trisolino, Pamela | Cristofanelli and Trisolino (2022) |
| CMN | ICOS ATC CH4 Release | Cristofanelli, Paolo and Trisolino, Pamela | Cristofanelli and Trisolino (2021) |
| GAT | ICOS ATC NRT CH4 growing time series | Kubistin, Dagmar and Plaß-Dülmer, Christian and Kneuer, Tobias and Lindauer, Matthias and Müller-Williams, Jennifer | Kubistin et al. (2022a) |
| GAT | ICOS ATC CH4 Release | Kubistin, Dagmar and Plaß-Dülmer, Christian and Arnold, Sabrina and Lindauer, Matthias and Müller-Williams, Jennifer and Schumacher, Marcus | Kubistin et al. (2021c) |
| HPB | ICOS ATC NRT CH4 growing time series | Kubistin, Dagmar and Plaß-Dülmer, Christian and Kneuer, Tobias and Lindauer, Matthias and Müller-Williams, Jennifer | Kubistin et al. (2022b) |
| HPB | ICOS ATC CH4 Release | Kubistin, Dagmar and Plaß-Dülmer, Christian and Arnold, Sabrina and Lindauer, Matthias and Müller-Williams, Jennifer and Schumacher, Marcus | Kubistin et al. (2021d) |
| HTM | ICOS ATC NRT CH4 growing time series | Heliasz, Michal and Biermann, Tobias | Heliasz and Biermann (2022) |
| HTM | ICOS ATC CH4 Release | Heliasz, Michal and Biermann, Tobias | Heliasz and Biermann (2021) |
| IPR | ICOS ATC NRT CH4 growing time series | Manca, Giovanni | Manca (2022) |
| IPR | ICOS ATC CH4 Release | Bergamaschi, Peter and Manca, Giovanni | Bergamaschi and Manca (2021) |
| JFJ | ICOS ATC NRT CH4 growing time series | Emmenegger, Lukas and Leuenberger, Markus and Steinbacher, Martin | Emmenegger et al. (2022) |
| JFJ | ICOS ATC CH4 Release | Emmenegger, Lukas and Leuenberger, Markus and Steinbacher, Martin | Emmenegger et al. (2021) |
| KIT | ICOS ATC NRT CH4 growing time series | Kubistin, Dagmar and Plaß-Dülmer, Christian and Kneuer, Tobias and Lindauer, Matthias and Müller-Williams, Jennifer | Kubistin et al. (2022c) |
| KIT | ICOS ATC CH4 Release | Kubistin, Dagmar and Plaß-Dülmer, Christian and Arnold, Sabrina and Lindauer, Matthias and Müller-Williams, Jennifer and Schumacher, Marcus | Kubistin et al. (2021e) |
| KRE | ICOS ATC NRT CH4 growing time series | Marek, Michal V. and Vítková, Gabriela and Komínková, Kateřina | Marek et al. (2022) |
| KRE | ICOS ATC CH4 Release | Marek, Michal V. and Vítková, Gabriela and Komínková, Kateřina | Marek et al. (2021) |
| LIN | ICOS ATC NRT CH4 growing time series | Kubistin, Dagmar and Plaß-Dülmer, Christian and Kneuer, Tobias and Lindauer, Matthias and Müller-Williams, Jennifer | Kubistin et al. (2022d) |
| LIN | ICOS ATC CH4 Release | Kubistin, Dagmar and Plaß-Dülmer, Christian and Arnold, Sabrina and Lindauer, Matthias and Müller-Williams, Jennifer and Schumacher, Marcus | Kubistin et al. (2021f) |
| LMP | ICOS ATC NRT CH4 growing time series | di Sarra, Alcide and Piacentino, Salvatore | di Sarra and Piacentino (2022) |
| LMP | ICOS ATC CH4 Release | di Sarra, Alcide and Piacentino, Salvatore | di Sarra and Piacentino (2021) |
| LUT | ICOS ATC NRT CH4 growing time series | Chen, Huilin and Scheeren, Bert | Chen and Scheeren (2022) |
| LUT | ICOS ATC CH4 Release | Chen, Huilin and Scheeren, Bert | Chen and Scheeren (2021) |
| NOR | ICOS ATC NRT CH4 growing time series | Lehner, Irene and Mölder, Meelis | Lehner and Mölder (2022) |
| NOR | ICOS ATC CH4 Release | Lehner, Irene and Mölder, Meelis | Lehner and Mölder (2021) |
| OPE | ICOS ATC NRT CH4 growing time series | Ramonet, Michel and Conil, Sébastien and Delmotte, Marc and Laurent, Olivier and Lopez, Morgan | Ramonet et al. (2022a) |
| OPE | ICOS ATC CH4 Release | Ramonet, Michel and Conil, Sébastien and Delmotte, Marc and Laurent, Olivier | Ramonet et al. (2021a) |
| OXK | ICOS ATC NRT CH4 growing time series | Kubistin, Dagmar and Plaß-Dülmer, Christian and Kneuer, Tobias and Lindauer, Matthias and Müller-Williams, Jennifer | Kubistin et al. (2022e) |
| OXK | ICOS ATC CH4 Release | Kubistin, Dagmar and Plaß-Dülmer, Christian and Arnold, Sabrina and Lindauer, Matthias and Müller-Williams, Jennifer | Kubistin et al. (2021a) |
| PAL | ICOS ATC NRT CH4 growing time series | Hatakka, Juha | Hatakka (2022) |
| PAL | ICOS ATC CH4 Release | Hatakka, Juha | Hatakka (2021) |
| PUI | ICOS ATC NRT CH4 growing time series | Lehtinen, Kari and Leskinen, Ari | Lehtinen and Leskinen (2022) |
| PUY | ICOS ATC NRT CH4 growing time series | Colomb, Aurélie and Ramonet, Michel and Yver-Kwok, Camille and Delmotte, Marc and Lopez, Morgan and Pichon, Jean-Marc | Colomb et al. (2022) |
| PUY | ICOS ATC CH4 Release | Colomb, Aurélie and Ramonet, Michel and Yver-Kwok, Camille and Delmotte, Marc and Pichon, Jean-Marc | Colomb et al. (2021) |
| RUN | ICOS ATC NRT CH4 growing time series | De Mazière, Martine and Sha, Mahesh Kumar and Ramonet, Michel | De Mazière et al. (2022b) |
| RUN | ICOS ATC CH4 Release | De Mazière, Martine and Sha, Mahesh Kumar and Ramonet, Michel | De Mazière et al. (2021) |
| SAC | ICOS ATC NRT CH4 growing time series | Ramonet, Michel and Delmotte, Marc and Lopez, Morgan | Ramonet et al. (2022b) |
| SAC | ICOS ATC CH4 Release | Ramonet, Michel and Delmotte, Marc | Ramonet and Delmotte (2021) |
| SMR | ICOS ATC NRT CH4 growing time series | Mammarella, Ivan | Mammarella (2022) |
| SMR | ICOS ATC CH4 Release | Levula, Janne and Mammarella, Ivan | Levula and Mammarella (2021) |
| STE | ICOS ATC NRT CH4 growing time series | Kubistin, Dagmar and Plaß-Dülmer, Christian and Kneuer, Tobias and Lindauer, Matthias and Müller-Williams, Jennifer | Kubistin et al. (2022f) |
| STE | ICOS ATC CH4 Release | Kubistin, Dagmar and Plaß-Dülmer, Christian and Arnold, Sabrina and Lindauer, Matthias and Müller-Williams, Jennifer | Kubistin et al. (2021b) |
| SVB | ICOS ATC NRT CH4 growing time series | Smith, Paul and Marklund, Per | Smith and Marklund (2022) |
| SVB | ICOS ATC CH4 Release | Marklund, Per and Ottosson-Löfvenius, Mikaell and Smith, Paul | Marklund et al. (2021) |
| TOH | ICOS ATC NRT CH4 growing time series | Kubistin, Dagmar and Plaß-Dülmer, Christian and Kneuer, Tobias and Lindauer, Matthias and Müller-Williams, Jennifer | Kubistin et al. (2022g) |
| TOH | ICOS ATC CH4 Release | Kubistin, Dagmar and Plaß-Dülmer, Christian and Arnold, Sabrina and Lindauer, Matthias and Müller-Williams, Jennifer and Schumacher, Marcus | Kubistin et al. (2021g) |
| TRN | ICOS ATC NRT CH4 growing time series | Ramonet, Michel and Lopez, Morgan and Delmotte, Marc | Ramonet et al. (2022c) |
| TRN | ICOS ATC CH4 Release | Ramonet, Michel and Lopez, Morgan and Delmotte, Marc | Ramonet et al. (2021b) |
| UTO | ICOS ATC NRT CH4 growing time series | Hatakka, Juha and Laurila, Tuomas | Hatakka and Laurila (2022) |
| UTO | ICOS ATC CH4 Release | Laurila, Tuomas | Laurila (2021) |
| ZEP | ICOS ATC NRT CH4 growing time series | Lund Myhre, Cathrine and Platt, Stephen Matthew and Hermansen, Ove and Lunder, Chris | Lund Myhre et al. (2022) |
| ZEP | ICOS ATC CH4 Release | Lund Myhre, Cathrine and Platt, Stephen Matthew and Hermansen, Ove and Lunder, Chris | Lund Myhre et al. (2021) |



**Table A3.** The TCCON sites used for evaluation.

| ID | Station Name, Location | Latitude | Longitude | References |
|----|------------------------|----------|-----------|------------|
| br | Bremen, Germany | 53.1°N | 8.85°E | Notholt et al. (2022) |
| bu | Burgos, Philippines | 18.53°N | 120.65°E | Morino et al. (2022c) |
| ci | California Institute of Technology, USA | 34.14°N | 118.13°W | Wennberg et al. (2022b) |
| db | Darwin, Australia | 12.46°S | 130.89°E | Deutscher et al. (2023b) |
| df | Armstrong Flight Research Center, USA | 34.96°N | 117.88°W | Iraci et al. (2022) |
| et | East Trout Lake, Canada | 54.36°N | 104.99°W | Wunch et al. (2002) |
| eu | Eureka, Canada | 80.05°N | 86.42°W | Strong et al. (2022) |
| gm | Garmisch, Germany | 47.48°N | 11.06°E | Sussmann and Rettinger (2017) |
| hf | Hefei, China | 31.9°N | 117.17°E | Liu et al. (2023) |
| iz | Izaña, Spain | 28.3°N | 16.5°W | Blumenstock et al. (2017) |
| js | Saga, Japan | 33.24°N | 130.29°E | Shiomi et al. (2022) |
| ka | Karlsruhe, Germany | 49.1°N | 8.44°E | Hase et al. (2022) |
| ll | Lauder, New Zealand | 45.04°S | 169.68°E | Sherlock et al. (2022) |
| ni | Nicosia, Cyprus | 35.14°N | 33.38°E | Petri et al. (2023) |
| ny | Ny Ålesund, Norway | 78.92°N | 11.92°E | Buschmann et al. (2022) |
| oc | Lamont, USA | 36.6°N | 97.49°W | Wennberg et al. (2022c) |
| or | Orleans, France | 47.97°N | 2.11°E | Warneke et al. (2022) |
| pa | Park Falls, USA | 45.94°N | 90.27°W | Wennberg et al. (2022a) |
| pr | Paris, France | 48.85°N | 2.36°E | Te et al. (2022) |
| ra | Reunion Island, France | 20.9°S | 55.49°E | De Mazière et al. (2022a) |
| rj | Rikubetsu, Japan | 43.46°N | 143.77°E | Morino et al. (2022a) |
| so | Sodankylä, Finland | 67.37°N | 26.63°E | Kivi et al. ((2022) |
| tk | Tsukuba, Japan | 36.05°N | 140.12°E | Morino et al. (2022b) |
| wg | Wollongong, Australia | 34.41°S | 150.88°E | Deutscher et al. (2023a) |
| xh | Xianghe, China | 39.8°N | 116.96°E | Zhou et al. (2022) |



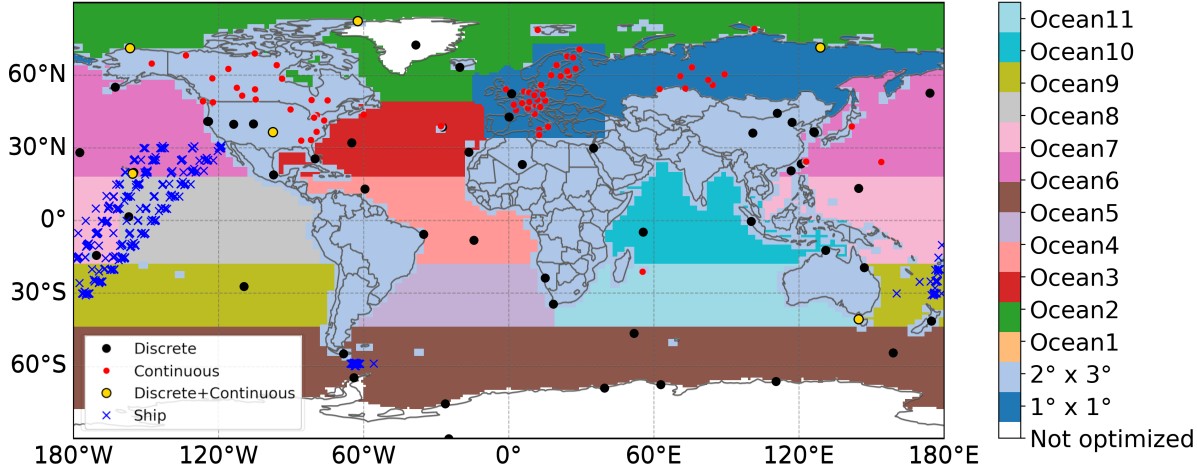

**Figure A1.** Location of ground-based surface observations assimilated in InvSURF (dots and x-marks), and optimization regions used in CTE-CH$_4$ (background colours). Over land areas, the fluxes were optimized grid-wise, and ocean region-wise.

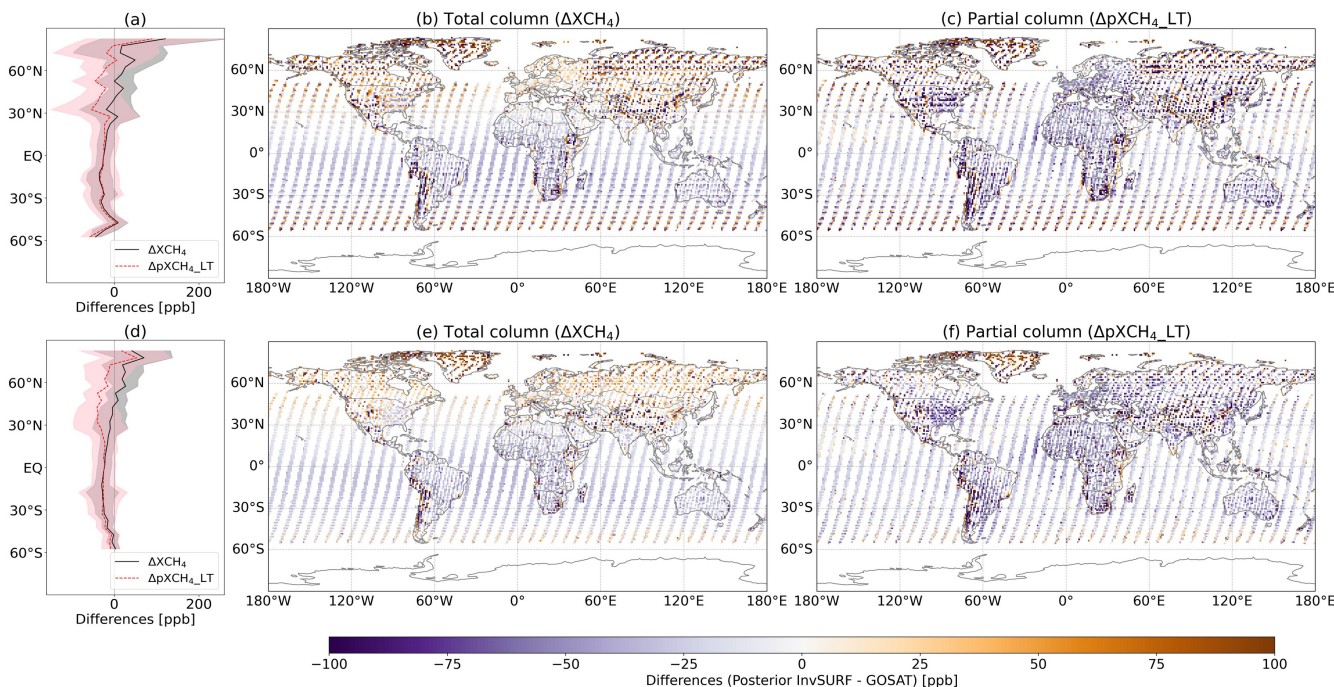

**Figure A2.** Mean differences between prior mole fractions and JAXA/GOSAT retrievals, averaged over 2016–2019. (a) and (d) five degree latitudinal means (solid line) with standard deviation (shaded area), and (b), (c), (e) and (f) 1° × 1° grid means. (a), (b) and (c) were calculated using TM5 with 1° × 1° zoom over Europe and 1° × 1° globally, and (d), (e) and (f) using 2° × 3° (latitude × longitude) globally. Positive values indicate posterior InvSURF mole fractions being higher than the GOSAT retrievals.



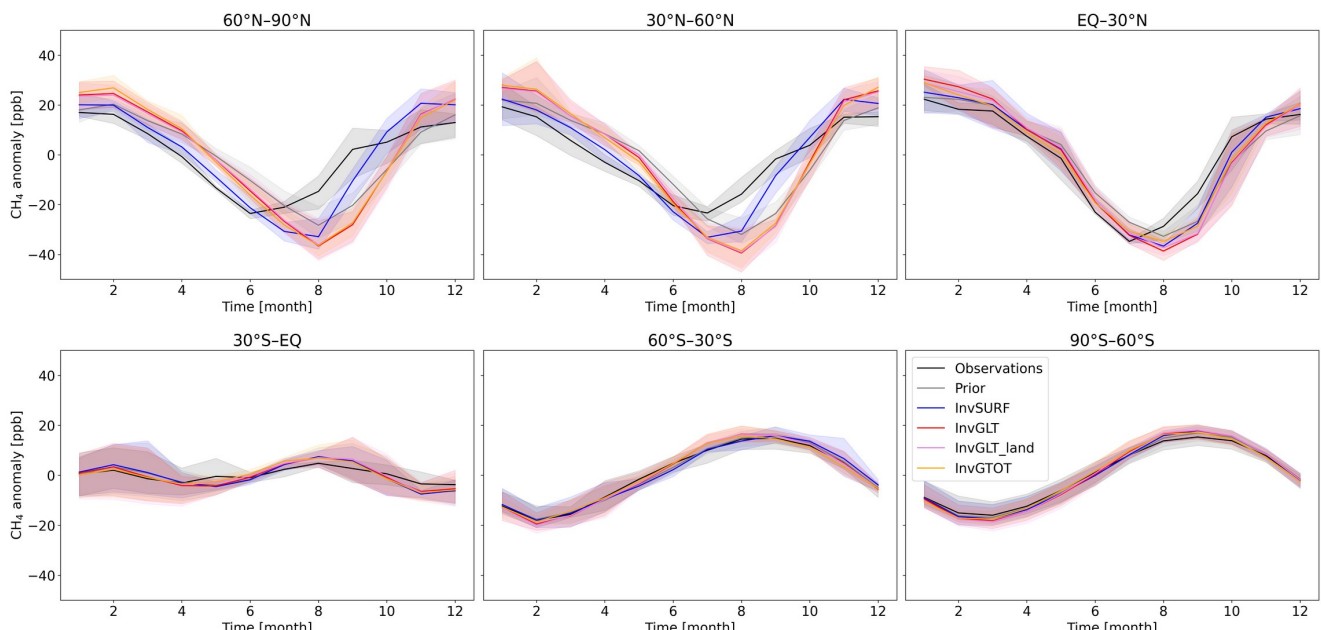

**Figure A3.** Detrended monthly average mole fractions at surface stations, assimilated in InvSURF. The data is averaged from 2016–2019 and at 30° latitudinal bands. Shaded areas are minimum and maximum of detrended monthly values within 2016–2019.

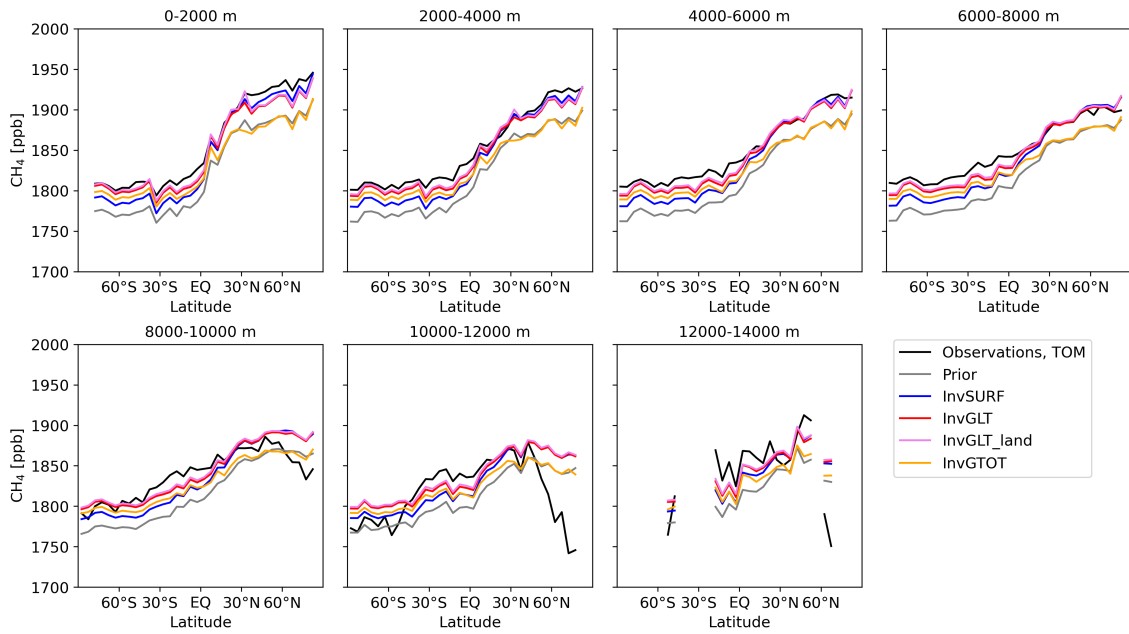

**Figure A4.** Mean atmospheric $CH_4$ at location and time of aircraft measurements from Atmospheric Tomography Mission, averaged over five degree latitude bands and 2000 m altitude bands above sea level during 2016–2019.





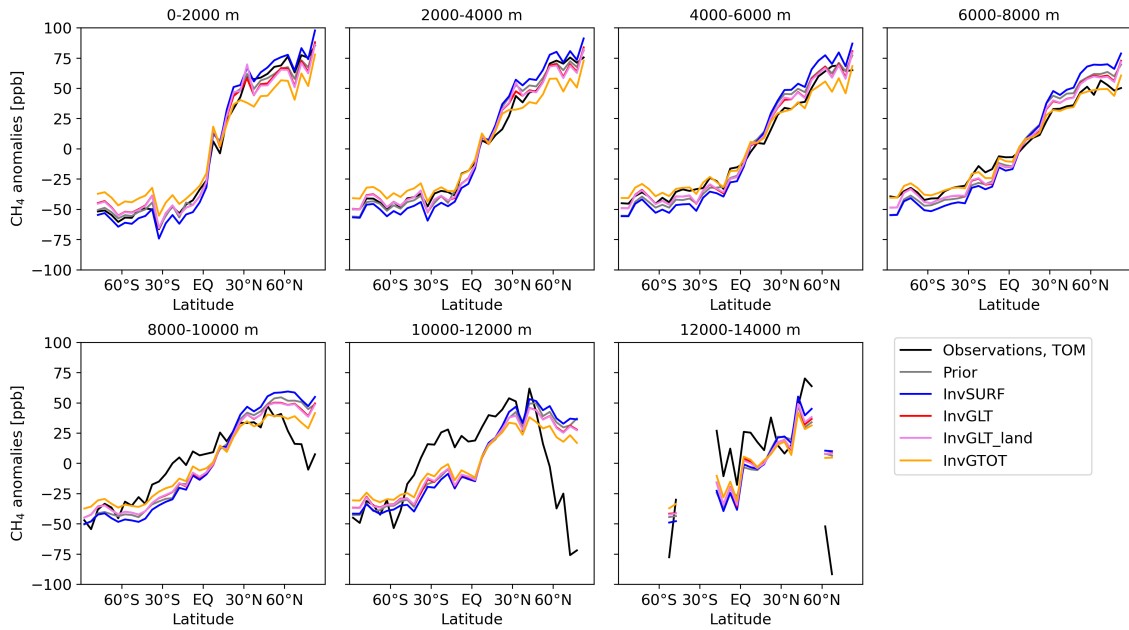

**Figure A5.** As in A4, but means were subtracted at each altitude bands.

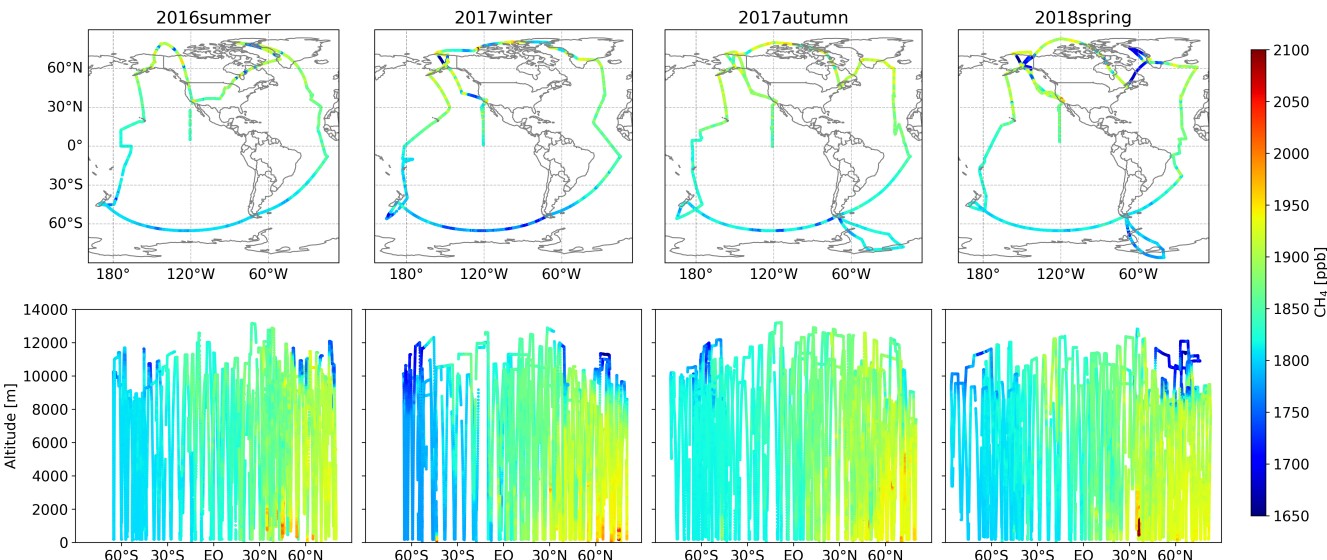

**Figure A6.** Observed CH$_4$ mole fractions from aircraft measurements of Atmospheric Tomography Mission during the study period.




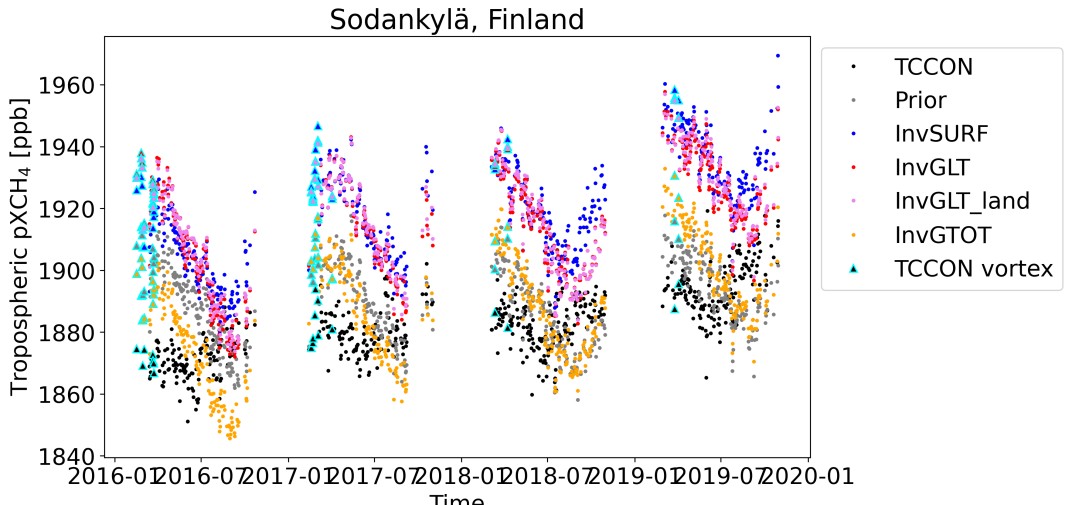

**Figure A7.** Daily averaged tropospheric partial columns at the Sodankylä TCCON station, Finland.

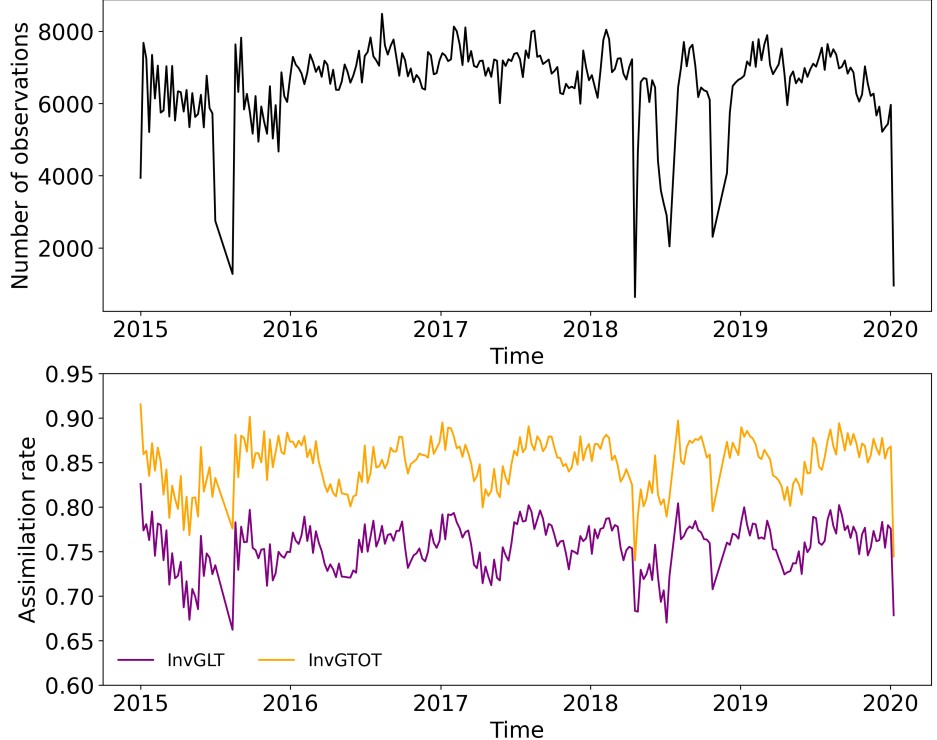

**Figure A8.** Global total number of JAXA/GOSAT observations per week and assimilation rates in GOSAT inversions. The x-axis is weeks of optimization steps, and not the actual time of observation.



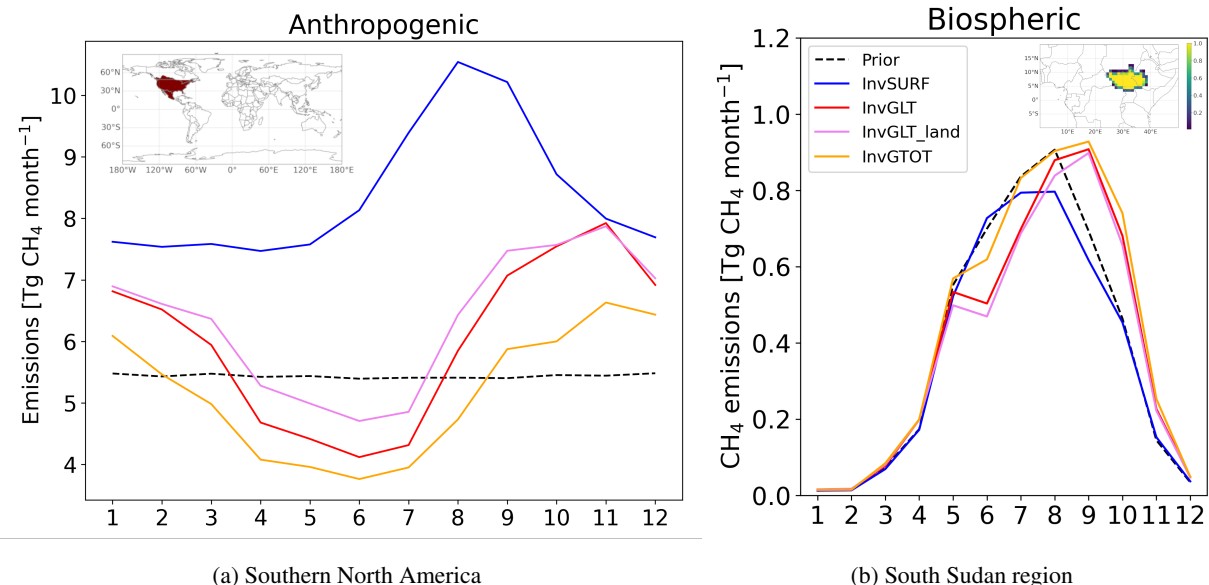

(a) Southern North America        (b) South Sudan region

**Figure A9.** Average monthly $CH_4$ emissions in (a) southern North America and (b) South Sudan regions. The maps illustrate aggregated areas.

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
