# Peer review of "Global CH4 Fluxes Derived from JAXA/GOSAT Lower Tropospheric Partial Column Data and the CTE-CH4 Atmospheric Inverse Model"

_EGUsphere, 2025_

## Referee Comment (RC1)

The manuscript by Tsuruta et al., titled "Global CH4 Fluxes Derived from JAXA/GOSAT Lower Tropospheric Partial Column Data and the CTE-CH4 Atmospheric Inverse Model," explores, for the first time, the potential of the new lower tropospheric partial column, comparing it to several other inversion setups. In general, the authors propose useful concepts for the effective use of partial columns to constrain surface CH4 emissions, which is the second most important greenhouse gas (GHG). The authors highlight the following main findings: 1) "the advantages of JAXA/GOSAT lower tropospheric partial column retrievals in estimating global and regional CH4 budgets", 2) "lower tropospheric partial column data possibly reduce global emission uncertainty," and 3) "highlighted the importance of transport model resolution in estimation of total and partial column data, indicating the need for high resolution transport models in satellite-driven inversions". The overall structure of the article is well-organized and balanced. The methods and techniques employed are appropriate. The manuscript is scientifically sound, references previous literature appropriately. At the same time, it also highlights critical issues that must be addressed to ensure the accuracy and reliability of its results before the manuscript publication.

**Major comments**

1. Unfortunately, the main results are inadequately supported in the manuscript text:

1.1. The test results are too evident and speculoos: InvSURF consistently outperforms in nearly all comparisons, including those with ground-based observations, demonstrating its superior performance in a broad range of cases. In contrast, InvGTOT proves to be more effective when compared to TCCON, as shown in Table 1. While the text emphasizes the benefits of InvGLT and InvGLT_land, these advantages appear to be overstated and are not substantiated by the observed results. The claimed improvements from InvGLT and InvGLT_land are not as pronounced or significant as suggested, and the data do not provide clear evidence to support the substantial benefits that the text implies.

1.2. The reduction in emission uncertainty is not adequately supported by the data or analysis presented. The discussion lacks sufficient evidence to substantiate the claimed improvements in uncertainty reduction. Furthermore, the sentence on lines 323-326: "Within the GOSAT inversions, the number of assimilated data were higher in InvGTOT than InvGLT (Fig. A8), and the observational uncertainty and rejection thresholds were the same in both inversions. This indicates that the lower uncertainty in InvGLT was not simply due to the number of assimilated observations" introduces

significant confusion, as it directly contradicts the fundamental principles of satellite inversion. Rayner and O'Brien (GRL, 2001) proved that a large number of satellite observations, even with moderate precision, can effectively constrain the global greenhouse gas (GHG) budget, a concept that has been further explored and validated in a variety of studies. The assertion that lower uncertainty in InvGLT is not attributable to the number of assimilated observations appears to be inconsistent with well-established principles of satellite data assimilation. Moreover, there are several critical questions that remain unaddressed and must be clarified for a more thorough understanding of the results. For instance, why is the maximum uncertainty reduction rate observed over ocean grids (Figure 6)? Additionally, why is there no significant difference in the uncertainty reduction between InvGLT and InvGLT_land? Given that InvGLT_land should involve a significantly different number of assimilated points due to the exclusion of oceanic regions, one would expect to observe clear discrepancies in the uncertainty reduction rates between these two inversion types. These unanswered questions call for further clarification and analysis to better support the conclusions regarding emission uncertainty reduction.

1.3 The effect of higher horizontal resolution on atmospheric models is well-documented and extensively discussed in the literature, as estimated in numerous studies (e.g., Houweling et al., ACP, 2004; Patra et al., ACP, 2011). These studies suggest that increased resolution leads to more accurate representation of atmospheric processes, with finer grids providing a more detailed and reliable depiction of the distribution of greenhouse gases, including methane. However, in this particular context, Figure A2 presents a particularly puzzling result that warrants further explanation. The figure demonstrates a clear discrepancy in the impact of grid resolution on the modeled data, as the difference calculated using the course 2° × 3° grid (Fig. A2d) is unexpectedly much smaller than the difference calculated using the finer 1° × 1° grid (Fig. A2a). This inconsistency in the figure requires clarification, as it challenges the well-established understanding that higher resolution typically enhances model accuracy. It also raises important questions about the potential influence of other factors, such as the observational data or the specific methodological approach used to process the results.

2. This study fails to account for averaging kernels, which are crucial for accurately comparing observed and modeled xCH4 values due to the inherent variability in the sensitivity of GOSAT instruments to vertical profiles of CH4. Averaging kernels represent instrumental sensitivity at different altitudes, and they play a key role in converting the observed column-integrated measurements into more precise information about the distribution of CH4 in the atmosphere. Without properly accounting for averaging kernels,

the comparison between the observed and modeled data could be misleading, as it would fail to fully capture the nuances of the instrument's sensitivity. Given that averaging kernels are now available in the latest version of the JAXA/GOSAT partial column product (v3.0), I strongly urge that all calculations in this study be revisited using the updated product. This would ensure that the analysis accounts for the improved accuracy and sensitivity provided by these kernels, which are crucial for obtaining more reliable results in satellite-based inverse modeling of methane emissions.

3. The definitions of parameters such as pXCH4_LT, pXCH4_LTmodel, and other partial column parameters utilized in this study are overly complex and may hinder the reader's understanding of the conceptual framework being presented. These definitions are essential for interpreting the study's methodology, but their complexity could lead to confusion for those not intimately familiar with the technical details. To improve clarity and facilitate a better understanding of the work, I strongly recommend the inclusion of a schematic plot that visually illustrates the concept of these parameters. A schematic diagram would provide a clear, step-by-step representation of how these parameters are defined and calculated, thereby enhancing the transparency of the study. This addition would greatly aid readers in comprehending the methodology and the relationships between the different variables involved.

**Minor comments:**

P2, L18: from Table 2. In https://www.esrl.noaa.gov/gmd/aggi/aggi.html, Global Radiative Forcing (W m$^{-2}$) is 0.561 in 2022 and 0.565 in 2023. Please clarify.

P3, L50: please consider correction: "before performing satellite inversions" -> "before performing satellite-based inversions"

P5, L134: Looks confusing, please revise: "Vertical interpolation was not applied, leading to potential biases."

L181: "the stratospheric partial column of $\mathrm{XCH_4}_{\mathrm{TCCON}}^{\mathrm{tropo}}$" the tropo index confusing in this context.

P10, Table 1: "Bias, root mean squared error (RSME) and Pearson's correlation against observations at surface ground-based stations assimilated in InvSURF, ATom aircraft measurements, and TCCON data." It seems other inversion cases should be added in the table description.

P11, Figure 3: It seems that the figure subtitles should be above panels.

---

## Author Comment (AC1)

**Replies to reviewers**

In the followings, reviewer's comments are in blue, reply in **black** and modified/added sentences in ***black italic***. The line and figure numbers in the answers are the ones in the revised manuscript. Please note that Figure 1, Figure A3 and Section 4.1 were added, and therefore, the figure and section numbers were changed accordingly.

**Reviewer 1**

The manuscript by Tsuruta et al., titled "Global CH4 Fluxes Derived from JAXA/GOSAT Lower Tropospheric Partial Column Data and the CTE-CH4 Atmospheric Inverse Model," explores, for the first time, the potential of the new lower tropospheric partial column, comparing it to several other inversion setups. In general, the authors propose useful concepts for the effective use of partial columns to constrain surface CH4 emissions, which is the second most important greenhouse gas (GHG). The authors highlight the following main findings: 1) "the advantages of JAXA/GOSAT lower tropospheric partial column retrievals in estimating global and regional CH4 budgets", 2) "lower tropospheric partial column data possibly reduce global emission uncertainty," and 3) "highlighted the importance of transport model resolution in estimation of total and partial column data, indicating the need for high resolution transport models in satellite-driven inversions". The overall structure of the article is well-organized and balanced. The methods and techniques employed are appropriate. The manuscript is scientifically sound, references previous literature appropriately. At the same time, it also highlights critical issues that must be addressed to ensure the accuracy and reliability of its results before the manuscript publication. Major comments 1.

Thank you very much for the review and valuable comments.

Unfortunately, the main results are inadequately supported in the manuscript text:

1.1. The test results are too evident and speculoos: InvSURF consistently outperforms in nearly all comparisons, including those with ground-based observations, demonstrating its superior performance in a broad range of cases. In contrast, InvGTOT proves to be more effective when compared to TCCON, as shown in Table 1. While the text emphasizes the benefits of InvGLT and InvGLT_land, these advantages appear to be overstated and are not substantiated by the observed results. The claimed improvements from InvGLT and InvGLT_land are not as pronounced or significant as suggested, and the data do not provide clear evidence to support the substantial benefits that the text implies.

We estimate surface fluxes, and as atmospheric concentrations near surface is more sensitive to the surface fluxes, agreement with the surface data (ground-based data and aircraft data in the lower troposphere), are better indicators of how well we constrain the surface fluxes. From this point of view, InvSURF, InvGLT and InvGLT_land performs better than InvGTOT. This is why we concluded that the flux estimates using the partial column data is potentially more reliable than those using total column data. In addition, as the reviewer points out, InvSURF outperformed, indicating that comparing flux estimates to InvSURF is a key metric for assessing the reliability of the results. Especially in the Northern Hemisphere temperate regions where surface data is relatively extensive, we think agreement with

InvSURF is important. In this respect, InvGLT and InvGLT_land agreed to InvSURF better than InvGTOT. Therefore, overall, we think improvements in InvGLT and InvGLT_land compared to InvGTOT can be justified and highlighted as key results.

In the comparison with ATom data above surface and tropospheric and total column TCCON data, we aim to evaluate model performance with focus on long-range transport, tropopause mixing and atmospheric chemistry. These are important to take into account when assimilating total column satellite data, while surface CH4 and partial column data are less sensitive to these transport modelling errors. To clarify this point, we added the following text in Section 3.3 L288-291:

> *Therefore, the evaluation again the TCCON data provides limited information about how well the inversions constrained the surface fluxes but focuses more on the model performance regarding long-range transport, tropopause mixing and atmospheric chemistry, which are important to take into account when assimilating total column satellite data.*

1.2. The reduction in emission uncertainty is not adequately supported by the data or analysis presented. The discussion lacks sufficient evidence to substantiate the claimed improvements in uncertainty reduction. Furthermore, the sentence on lines 323-326: "Within the GOSAT inversions, the number of assimilated data were higher in InvGTOT than InvGLT (Fig. A8), and the observational uncertainty and rejection thresholds were the same in both inversions. This indicates that the lower uncertainty in InvGLT was not simply due to the number of assimilated observations" introduces significant confusion, as it directly contradicts the fundamental principles of satellite inversion. Rayner and O'Brien (GRL, 2001) proved that a large number of satellite observations, even with moderate precision, can effectively constrain the global greenhouse gas (GHG) budget, a concept that has been further explored and validated in a variety of studies. The assertion that lower uncertainty in InvGLT is not attributable to the number of assimilated observations appears to be inconsistent with well-established principles of satellite data assimilation. Moreover, there are several critical questions that remain unaddressed and must be clarified for a more thorough understanding of the results. For instance, why is the maximum uncertainty reduction rate observed over ocean grids (Figure 6)? Additionally, why is there no significant difference in the uncertainty reduction between InvGLT and InvGLT_land? Given that InvGLT_land should involve a significantly different number of assimilated points due to the exclusion of oceanic regions, one would expect to observe clear discrepancies in the uncertainty reduction rates between these two inversion types. These unanswered questions call for further clarification and analysis to better support the conclusions regarding emission uncertainty reduction.

Thank you for the critical comments. We agree with the reviewer that analysis/discussion on uncertainty focused on number of assimilated data and did not sufficiently answer the questions raised.

Regarding the differences in the uncertainty reduction rates between InvGTOT and InvGLT, we think this is due to sensitivities of the data to surface fluxes. Due to its nature of measured quantity, total columns have less sensitivity to surface fluxes than ground-based data or lower tropospheric partial column data, i.e. the total column data would have less power to constraint on surface fluxes. Therefore, just having a larger number of observations assimilated may not lead to higher uncertainty reduction rates. To clarify this point, we modified/added the following sentences in Section 3.5 L349-353:

> *This indicates that the lower uncertainty in InvGLT was not simply due to number of assimilated observations but also related to sensitivity of the data to surface fluxes. Due to the nature of the retrieved quantity, total columns have less sensitivity to surface fluxes than ground-based data or lower tropospheric partial column data, i.e. the total column data would have less power to constraint on surface fluxes. Therefore, the larger number of assimilated observations in InvGTOT did not necessary lead to higher uncertainty reduction rates than in InvGLT.*

Regarding the uncertainty reduction rates over ocean, i.e. difference in the uncertainty reduction between InvGLT and InvGLT_land, we must note that the rate ($\sigma_{prior}$ − $\sigma_{posterior}$)/$\sigma_{prior}$) depend on prior emissions and uncertainties, which are sensitive to small changes when $\sigma_{prior}$ are small. In this study, we assigned smaller prior uncertainty ($\sigma_{prior}$) over ocean (20% of emissions) compared to land (80%). The emissions over ocean were also much smaller than those over land (see e.g. Fig. 6). This ended up with prior uncertainty of ~$10^{-22}$ mol m$^{-2}$ s$^{-1}$ over ocean, which is approximately $10^{12}$ smaller than those over land. This is the main reason why uncertainty reduction rates over ocean became large, and did not vary much between the inversions even though the number of assimilated observations varied significantly. To clarify this point, we have added the following in Section 3.5 L372-379:

> *The uncertainty reduction rates over the ocean were relatively large compared to those over land (Fig. 7). This is mainly due to the assigned prior uncertainty, which was significantly smaller over the ocean; the emissions were low (Fig. 6) and prior uncertainty was set to 20% over the ocean (see Section 2.1). This has led to prior uncertainty of ~$10^{-22}$ mol m$^{-2}$ s$^{-1}$ over the ocean, which is approximately $10^{12}$ smaller than those over land. The uncertainty reduction rates are sensitive to small changes when the prior uncertainties are small, and therefore, they became larger over the ocean, although the absolute changes in the uncertainties were small.*

1.3 The effect of higher horizontal resolution on atmospheric models is well documented and extensively discussed in the literature, as estimated in numerous studies (e.g., Houweling et al., ACP, 2004; Patra et al., ACP, 2011). These studies suggest that increased resolution leads to more accurate representation of atmospheric processes, with finer grids providing a more detailed and reliable depiction of the distribution of greenhouse gases, including methane. However, in this particular context, Figure A2 presents a particularly puzzling result that warrants further explanation. The figure demonstrates a clear discrepancy in the impact of grid resolution on the modeled data, as the difference calculated using the course 2° × 3° grid (Fig. A2d) is unexpectedly much smaller than the difference calculated using the finer 1° × 1° grid (Fig. A2a). This inconsistency in the figure requires clarification, as it challenges the well-established understanding that higher resolution typically enhances model accuracy. It also raises important questions about the potential influence of other factors, such as the observational data or the specific methodological approach used to process the results.

We apologise for confusion. The figure caption was wrong. Fig. A2a is run with 1° × 1° zoom over Europe and 6° × 4° globally, so these results are consistent with the previous studies. The caption is corrected.

2. This study fails to account for averaging kernels, which are crucial for accurately comparing observed and modeled xCH4 values due to the inherent variability in the sensitivity of GOSAT instruments to vertical profiles of CH4. Averaging kernels represent instrumental sensitivity at different altitudes, and they play a key role in converting the observed column-integrated measurements into more precise information about the distribution of CH4 in the atmosphere. Without properly

accounting for averaging kernels, the comparison between the observed and modeled data could be misleading, as it would fail to fully capture the nuances of the instrument's sensitivity. Given that averaging kernels are now available in the latest version of the JAXA/GOSAT partial column product (v3.0), I strongly urge that all calculations in this study be revisited using the updated product. This would ensure that the analysis accounts for the improved accuracy and sensitivity provided by these kernels, which are crucial for obtaining more reliable results in satellitebased inverse modeling of methane emissions.

We agree that averaging kernel (AK) is important to take into account when possible. We have carried out an extra simulation by running TM5 forward with posterior InvSURF fluxes and compared with v3.0 data, taking into account averaging kernels and vertical interpolation (see the figure below, which is added as Fig. A3 in the revised manuscript). The largest differences in the comparisons against v2.0 and v3.0 were seen in the Northern Hemisphere (NH) temperate regions, where modelled XCH4 showed negative bias against the JAXA/GOSAT v3.0 data while it was positive against v2.0 data. This could possibly lead to larger emissions in the NH temperate regions using v3.0 data and may improve the agreement with the various observations from InvGTOT. For comparison to the lower tropospheric partial column data, ΔXCH4_LT is turned from positive to negative from v2.0 to v3.0 in regions which are e.g. highly elevated and northeastern part of China, which have large emissions. However, the large differences means that the observations are likely to be rejected during assimilation and would not constrain fluxes. Over the land regions with smaller biases, which data likely affect the flux results, we do not find significant changes from v2.0 to v3.0 in lower troposphere.

It is difficult to estimate how the comparison of inversion results between InvGTOT and InvGLT changes by using v3.0 data without properly running the inversions. However, despite different perspectives, this could again be considered as an advantage of using the partial column data. It is less sensitive to interpolation and AK than total columns, potentially providing more robust estimates from various transport models that differ in modelling long-range transport, tropopause mixing, and atmospheric chemistry. Nevertheless, rerunning all inversions and analysis with new data requires a considerable effort, and therefore, we hope the reviewer agrees to leave this for next paper.

We have added a new figure (Fig. A3 in the revised manuscript, see below) and the following new section to deepen the discussion about effect of averaging kernel and vertical interpolation:

*Section 4.1: Effect of vertical interpolation and averaging kernels*

*In this study, we did not apply vertical interpolation for simplicity or averaging kernels (AK) as JAXA/GOSAT v2.0 did not provide the information. However, we acknowledge that these are important to take into account when possible. With test simulations with TM5, we found that the latitudinal biases improved in ΔXCH4 when compared to JAXA/GOSAT v3.0 data when vertical interpolation and AK were applied (Fig. A3). In the NH temperate regions, the biases in ΔXCH4 turned from positive to negative and became similar to the biases in the tropics. Therefore, using v3.0 would possibly increase the CH4 flux estimates from the NH temperate regions in InvGTOT, aligning better with the inversions using surface and lower tropospheric partial column data. There are still positive biases in the high latitudes, although they are smaller than these compared to v2.0 data without interpolation and AK.*

*For lower tropospheric partial column, ΔpXCH4 _LT, the overall latitudinal biases did not change significantly by applying vertical interpolation and AK (Fig. A3). However, the large positive biases in highly elevated regions (e.g. Andean region in South America and Tibetan plateau in China and surroundings) and northeastern part of China, where CH4 emissions are large, turned to large negative biases. Nevertheless, large biases, both in positive and negative, means that the observations are likely to be rejected during assimilation and would not affect flux estimates. The ΔpXCH4 _LT over the ocean showed less latitudinal biases – the negative biases in the SH tropics turned positive. Overall, the robustness of modelling lower tropospheric partial column could be considered as an advantage over total column, having potential to provide more robust estimates regardless of differences in how transport models represent long-range transport, tropopause mixing, and atmospheric chemistry.*

[Figure]

**Figure A3.** (a)-(d) Mean differences between posterior InvSURF and JAXA/GOSAT retrievals averaged over 2016 and at 2° × 3° grids. Positive values indicate posterior InvSURF mole fractions being higher than the JAXA/GOSAT retrievals. (a) and (b) compared to v2.0, without averaging kernels and vertical interpolation. (c) and (d) compared to v3.0 with vertical interpolation and averaging kernels applied. (e) and (f) are comparisons of modelled values between those calculated with vertical interpolation and AK (i.e. posterior InvSURF values from (c) and (d)) and those without interpolation and AK (i.e. posterior InvSURF values from (a) and (b)), illustrating the effect of interpolation and AK directly. Positive values indicate higher mole fraction values with interpolation and AK. (a), (c), (e) are comparisons of total columns and (b), (d), (f) are of lower tropospheric partial columns.

3. The definitions of parameters such as pXCH4_LT, pXCH4_LTmodel, and other partial column parameters utilized in this study are overly complex and may hinder the reader's understanding of the conceptual framework being presented. These definitions are essential for interpreting the study's methodology, but their complexity could lead to confusion for those not intimately familiar with the technical details. To improve clarity and facilitate a better understanding of the work, I strongly recommend the inclusion of a schematic plot that visually illustrates the concept of these parameters. A schematic diagram would provide a clear, step-by-step representation of how these parameters are defined and calculated, thereby enhancing the transparency of the study. This addition would greatly aid readers in comprehending the methodology and the relationships between the different variables involved.

Thank you for the nice suggestion. We have created a schematic diagram and added as Figure 1.

[Figure]

**Figure 1.** A schematic figure illustrating how total column ($XCH_4$) and lower tropospheric partial column ($pXCH_4\_LT$) are calculated in the JAXA/GOSAT data and from TM5. JAXA/GOSAT retrieval algorithm uses information from both solar reflected light and thermal emissions to retrieve partial column $CH_4$ mole fractions. In JAXA/GOSAT there are five layers between the retrieved pressure ($sp_{GOSAT}^{ret}$) and 0.01 Pa, with two tropospheric and three stratospheric layers. The lower troposphere (LT) is defined as pressure levels between $sp_{GOSAT}^{ret}$ and $0.6 \times sp_{GOSAT}^{ret}$. In TM5, there are 25 model layers and $XCH_4^{model}$ is calculated using all layers, i.e. the pressure levels between $p_{j=0}^{model}$ and $p_{j=26}^{model}$. For calculation of $pXCH_4\_LT^{model}$, the minimum level (m) and maximum level (n) vary depending on $sp_{GOSAT}^{ret}$. $m = 0$ if $sp_{GOSAT}^{ret} > p_{j=0}^{model}$ and otherwise the maximum level at which $p_j^{model}$ exceeds $sp_{GOSAT}^{ret}$. n is the level at which $p_j^{model}$ reaches $0.6 \times sp_{GOSAT}^{ret}$.

Correspondingly, we have slightly modified the description of $pXCH_4\_LT^{model}$ calculation in Section 2.3.1 L136-139 as follows:

> *For calculating modelled lower tropospheric partial columns of methane, pXCH4 _LTmodel ,*
> *the layers were selected based on sp^ret_GOSAT (see also Fig. 1). The minimum layer is i = 0 if the*

*surface pressure in TM5 model is smaller than $sp^{ret}_{GOSAT}$ and otherwise the maximum layer at which TM5 model pressure exceeds the GOSAT-retrieved surface pressure. The maximum layer corresponded to the point where TM5 pressure reached 0.6 × $sp^{ret}_{GOSAT}$.*

Minor comments:

P2, L18: from Table 2. In https://www.esrl.noaa.gov/gmd/aggi/aggi.html, Global Radiative Forcing (W m-2 ) is 0.561 in 2022 and 0.565 in 2023. Please clarify.

Thank you for pointing out. The phrase is corrected as:

*a radiative forcing of 0.565 W m$^{-2}$ (for 2023;*

P3, L50: please consider correction: "before performing satellite inversions" -> "before performing satellite-based inversions"

The phrase is corrected accordingly.

P5, L134: Looks confusing, please revise: "Vertical interpolation was not applied, leading to potential biases."

We did not apply vertical interpolation, and therefore, modelled mixing ratio was calculated using thinner (in case $sp^{ret}_{GOSAT}$ > $sp^{model}$) or thicker (in case $sp^{ret}_{GOSAT}$ < $sp^{model}$) air mass from the lowest layer. For the uppermost layer, due to the selection method, the model nearly always contained thicker air mass than the retrievals. These likely cause some systematic biases. We have added this explanation in L139-142:

*Vertical interpolation was not applied for simplicity, and therefore, modelled mixing ratio was calculated using thinner (in case $sp^{ret}_{GOSAT}$ > $sp^{model}$) or thicker (in case $sp^{ret}_{GOSAT}$ < $sp^{model}$) air mass from the lowest layer. For the uppermost layer, due to the selection method, model nearly always contained thicker air mass than the retrievals. These likely lead to potential biases (see also Section 4.1).*

L181: "the stratospheric partial column of " the tropo index confusing in this context.

Thank you for pointing out. The "stratospheric" was wrong. The phrase is corrected as:

*the tropospheric partial column*

P10, Table 1: "Bias, root mean squared error (RSME) and Pearson's correlation against observations at surface ground-based stations assimilated in InvSURF, ATom aircraft measurements, and TCCON data." It seems other inversion cases should be added in the table description.

Here, InvSURF only was mentioned because that is only the inversion that used surface data for constraining fluxes. The comparisons were done for all inversions, as can be seen from the table, but for the comparison with surface ground-based stations, we selected the subset of data, which were assimilated in InvSURF. To clarify, we modified the caption as:

*…observations at surface ground-based stations (that were assimilated in InvSURF),…*

P11, Figure 3: It seems that the figure subtitles should be above panels.

Thank you for pointing out. The positions of subtitles are corrected as suggested.

**Reviewer 2**

General comments:

Since CH4 is one of the most important greenhouse gases in Earth's atmosphere, at lot of research effort is put into measuring its concentration either in situ or remotely. Those measurement can be used to derive methane fluxes at the surface using various inversion techniques. The authors use a novel methane retrieval from JAXA/GOSAT in an existing ensemble Kalman filter based inversion system. The main conclusion is that assimilating the lower tropospheric column from the novel retrieval is more consistent with the results from a retrieval using surface data than with a retrieval using total column data. This is an interesting conclusion in view of other synergistic retrievals using the SWIR and TIR wavelength bands from other instruments. Therefore, the paper presents an important contribution to the field and is worthy of prompt publication, after careful consideration of the comments presented below.

Thank you very much for the review and positive comments.

Specific comments

Section 2.1:

-) Some clarification of the horizontal resolutions would be appreciated. If the fluxes are optimized at 1x1 over Eurasia and 2x3 over other land areas, how does this work with a model resolution of 1x1 over Europe and 6x4 over the rest of the globe?

Our system can optimize fluxes at resolution independent of transport model resolution. However, we agree with the reviewer that using different resolutions can be questionable especially in the case when transport model resolution is lower than the flux optimization resolution because the atmospheric states may not be resolved in detail enough. We think this point can be justified considering the level of detail discussed in this paper, which is mostly focusing on large-scale fluxes. It is also our aim to increase transport model resolution in next study, and we will continue investigating and improving model resolutions (for both in terms of flux optimization and transport). The following sentence is added in L97-100:

> *The resolution is coarser than the flux optimization resolution outside Europe. We acknowledge that using different resolutions can be questionable because the atmospheric states may not be resolved in detail enough. However, this could be justified as the study mostly focus on large-scale fluxes.*

-) Why choose for 1x1 over Europe and 6x4 over the rest of the globe anyway? You focus on global results, so wouldn't it make more sense to use the same resolution everywhere? You did run the model on 1x1 and 3x2 outside Europe anyway for figure A2.

This setup was left from our previous studies (e.g. Thompson et al., 2021; Tenkanen et al., 2025), which has been focusing on European regions, although we agree that globally homogeneous resolution would have been more suitable for this study. In Fig. A2, we did global 3° x 2° resolution simulation without zoom region. We modified the text in L96-97 as:

TM5 was run at 1° × 1° over Europe and 6° × 4° globally, *following e.g. Thompson et al. (2021) and Tenkanen et al. (2025), ...*

-) It is mentioned that the atmospheric chemistry is the same as in Houweling (2014), which uses the OH distribution from Spivakovski (2000). That is also mentioned in line 365. But then please clarify the use of the ECHAM/MESSy1 model in line 98.

Sorry for the confusion and thank you for pointing out. Indeed, the OH fields were same as in Houweling (2014), which is based on Spivakovski (2000), but scaled by 0.92. For Cl and O($^1$D), we use the reaction rates calculated from ECHAM/MESSy1 model, which is different from the ones used in Houweling (2014). The texts in L101-104 are modified as:

*The OH concentrations were same as in Houweling et al. (2014), which is based on Spivakovsky et al. (2000) distribution but scaled globally by 0.92. For reactions with Cl and O($^1$D), the reaction rates pre-calculated from the ECHAM/MESSy1 model (Jöckel et al., 2006) were used.*

Section 2.3.1:

-) In lines 133-135, two major shortcomings in the current approach are mentioned: the lack of vertical interpolation and application of averaging kernels. That may be understandable for the AKs, since they are lacking the v2 version of the GOSAT product. However, more explanation on why vertical interpolation was not applied should be added to the text. And in the discussion, the possible effects of not including vertical interpolation or not applying the AK should be discussed.

The vertical interpolation was not done for simplicity. Following the reviewer's advice, we have carried out additional simulations with TM5 in forward mode and estimated XCH4 and pXCH4_LT by applying vertical interpolation and AK information from v3.0 (see the figure above in reply to reviewer 1 general comment 2, which is Fig. A3 in the revised manuscript).

We also have added discussion in Section 4.1 regarding the effect of vertical interpolation and AK:

*Section 4.1: Effect of vertical interpolation and averaging kernels*

*In this study, we did not apply vertical interpolation for simplicity or averaging kernels (AK) as JAXA/GOSAT v2.0 did not provide the information. However, we acknowledge that these are important to take into account when possible. With test simulations with TM5, we found that the latitudinal biases improved in ΔXCH4 when compared to JAXA/GOSAT v3.0 data when vertical interpolation and AK were applied (Fig. A3). In the NH temperate regions, the biases in ΔXCH4 turned from positive to negative and became similar to the biases in the tropics. Therefore, using v3.0 would possibly increase the CH4 flux estimates from the NH temperate regions in InvGTOT, aligning better with the inversions using surface and lower tropospheric partial column data. There are still positive biases in the high latitudes, although they are smaller than these compared to v2.0 data without interpolation and AK.*

*For lower tropospheric partial column, ΔpXCH4_LT, the overall latitudinal biases did not change significantly by applying vertical interpolation and AK (Fig. A3). However, the large positive biases in highly elevated regions (e.g. Andean region in South America and Tibetan plateau in China and surroundings) and northeastern part of China, where CH4 emissions are large, turned to large negative biases. Nevertheless, large biases, both in positive and negative,*

*means that the observations are likely to be rejected during assimilation and would not affect flux estimates. The ΔpXCH4 _LT over the ocean showed less latitudinal biases – the negative biases in the SH tropics turned positive. Overall, the robustness of modelling lower tropospheric partial column could be considered as an advantage over total column, having potential to provide more robust estimates regardless of differences in how transport models represent long-range transport, tropopause mixing, and atmospheric chemistry.*

-) In lines 139-141 (and again in line 203), the observational uncertainty is set to 30 or 50 ppb depending on the product and location, and the rejection threshold is set to two times those values. But if the uncertainty is set to a fixed value, then the observations will never be rejected. So some clarification on the use of the rejection threshold should be added to the text.

The fixed observation uncertainty is predefined as diagonal of R. The rejection threshold means that if differences between observed and prior-modelled mole fractions exceed the threshold, the observations are rejected and would not be used to constrain the fluxes. It is true that if rejection threshold is too large, the observations would never be rejected. However, in this study, the rejection rates were around 20% in the GOSAT inversions, i.e. around 80% of observations were assimilated to constrain the fluxes (Fig. A9). The following sentence is added in L149-151:

*The rejection thresholds discriminate the observations if differences between observed and prior-modelled mole fractions exceed the threshold, i.e. the observations are rejected and would not be used to constrain the fluxes.*

Section 2.4:

-) Since the CTE-CH4 implements the ensemble Kalman filter, it is surprising that the number of ensemble members is not mentioned in the text. According to the caption of figure 4, there are 500 ensemble members, but that may be a bit high. Please clarify.

Yes, we have used 500 members, based on experiments in Tsuruta et al. (2017). We have added the following sentence in L94-95:

*Regarding EnKF setups, we used an ensemble size of 500 members and optimization window of 7-days with lag of 5 following Tsuruta et al. (2017).*

-) Why are the four inversion experiments from this section selected? More specific, what is the reason for doing the InvGLT_land experiment?

This was to test in consideration of assimilating other satellite data, such as TROPOMI, which do not necessary provide ocean data from all retrieval products. Also, considering relatively small contribution of ocean fluxes to global total, compared to that of e.g. CO2, we wished to test if land fluxes would be constrained equally well without ocean data. We have added the following sentences in L212-215:

*The inversion without ocean data were tested in consideration of assimilating other satellite data, such as TROPOMI, which do not necessary provide ocean data from all retrieval products. Additionally, considering relatively small contribution of ocean fluxes to global total, compared to that of e.g. CO2, we examine if land fluxes would be constrained equally well without ocean data.*

Section 3.1:

-) Lines 218-221: Is it possible that not applying vertical interpolation (see section 2.3.1) or the coarse model resolution outside Europe also contribute to the observed biases? For example, the surface elevation of the model over mountainous regions such as western South America (i.e. Andes mountains) will be different from the surface elevation of the satellite measurements. This difference in elevation will affect the derived methane column. And how will the lack of AKs affect these biases?

As pointed by the reviewer, the coarse spatial resolution outside Europe contributes the biases. The large biases in regions over mountain regions are slightly reduced in some regions by increasing the horizontal resolution of TM5 (Fig. A2). However, the high biases still remain in regions such as Andes mountains in South America and Tibetan plateau in China and surroundings.

We carried out additional simulations with TM5 in forward mode and estimated XCH4 and pXCH4_LT by applying vertical interpolation and AK information from v3.0 (see Fig. A3 in the revised manuscript). For XCH4 the biases in the mountain regions are smaller when compared to v3.0. For pXCH4_LT the sign of biases has changed from positive to negative, but absolute magnitudes did not change significantly.

We have added the following sentences in L236-241:

> *The horizontal resolutions of the transport model also contribute to the biases in highly elevated areas. The additional simulation with TM5 showed that increasing resolution from 4° × 6° (latitude × longitude) to 2° × 3° decreased biases in mountain regions in Africa and Tibetan plateau, although the biases in regions such as Andes mountains in South America and edges of Tibetan plateau still remain (Fig. A2). We also acknowledge that the vertical interpolation and averaging kernel (AK) contribute to the biases in highly elevated areas, especially for XCH4 (Fig. A3) (see also Section 4.1).*

-) Line 230: Comparing figure A2 with with figure 1, it appears that the (SH) tropical bias deteriorates with higher resolution. If smaller biases over Europe are the result of a higher model resolution, then why is that not the case globally? I agree that the biases between Europe and Russia are smaller for 3x2 (A2, def), but this is not true for the inversion with the total column (A2, b) on global 1x1 resolution.

We apologies for confusion. The figure caption was wrong. Figure A2a is run with 1° × 1° zoom over Europe and 6° × 4° globally. The caption is corrected.

-) Line 232: ... smaller during NH winter than summer... However, if I look at Figure 2, I would say that the minimum is more in October than in NH winter.

The reviewer is true. L251 is modified as:

> *…smaller during NH autumn-winter than spring-summer.*

Section 3.2:

-) Figure 3: What are the "obs", surface values or total columns? I don't think that they are total columns, since the observations are "surface ground based and shipboard". But InvGTOT assimilated the total columns, so are you showing the surface concentration from that experiment here? In other words, are you comparing the same quantities here? Total column XCH4 will be lower than surface observations due to the shape of the CH4 profile. In the caption, please rephrase the line "For (a), the

data that were assimilated in InvSURF were used". But that inversion is shown in every plot, so that is meant by that line?

Here, the modelled estimates using prior or posterior fluxes from each inversion experiment are compared against the observations. The "observations" are (a) Surface stations, (b) ATom, 0–2000 m, (c) tropospheric partial column at TCCON stations and (d) total column at TCCON stations. In all cases, we run TM5 in the forward mode, using the posterior fluxes obtained from each inversion, to estimate posterior mole fractions for all observed quantities (either for a single altitude point or average of sub- or total columns). The following explanation is added in the caption:

> *The coloured lines are the posterior estimates from each inversion, which is estimated by running TM5 forward using posterior fluxes of the corresponding inversion simulation.*

-) Line 255: "… InvGTOT below 4000 meter …", but Fig 3b mentions 2000m. Please clarify.

Here, with "below 4000 m" we meant to include the altitude band of 0-2000 m as well, which is shown in Figure 3b. To avoid confusion, we modified the sentence as:

> … *InvGTOT at altitude bands 0-2000 and 2000-4000 meter*…

-) Line 259/260: I'm not sure that InvGTOT better captures the latitudinal gradient. InvGTOT still underestimates the observations in the 4000-8000 meter range. Please add some more explanation.

It is true that InGTOT still underestimates the observations, but the latitudinal gradient, i.e. as anomaly, is better captured. To clarify, we removed the reference to Fig. A5 (previously A4), which shows absolute differences, but we wish to highlight the anomalies, which are better illustrated in Fig. A6 (previously A5).

Technical corrections

line 7:    change was to were

Corrected.

line 55:    change inversion to inversions

Corrected.

line 59:    remove the first occurrence of thermal

Corrected.

line 66:    remove "in"

"in" is replaced by "of"

line 82 / eq. 1: Below this equation, you do not mention $x^b$, and in line 83/84 y and x are in italic, while in the equation they are in bold. And there's an error in the equation as well: the observation operator does not operate on $x^b$, but on x.

Thank you for pointing out. All are corrected.

line 110:    in general, I think the English in section 2.3.1 should be reviewed

The language is checked with a native speaker.

line 111/112:   Replace with "The JAXA/GOSAT retrieval algorithm is based on the Full Physics algorithm and is extended to use simultaneous both the 2-orthogonal SWIR and TIR signals."

Revised accordingly.

line 113:    replace "and use..." with "and uses..."

Corrected.

line 114/115:   Please rephrase the line "The an ... retrieval process".

The sentence is revised as:

> *The empirical orthogonal function (EOF) fitting is taken account in the retrieval process, where $XCO_2$, $XCH_4$ and $XH_2O$ are simultaneously retrieved with and solar-induced chlorophyll fluorescence (SIF) information.*

line 116/117:   Replace "Two layers ... are derived" with something like "CO2 and CH4 partial column-averaged concentrations are derived for two layers in the troposphere and three layers in the stratosphere. The H2O concentration is derived on 11 vertical layers."

Revised accordingly.

line 118:    Add "CO2 and CH4" to the start of the line: "The five CO2 and CH4 layers..."

Added.

line 143:    "examined" should be "justified"?

Modified accordingly.

line 144:    add a "d" to "compare"

Added.

line 228/229:   Please rephrase the line "This resulted...the shift".

line 233:    Remove "it" from "... biases it showed ..."

Removed.

line 256:    Replace "this" with "these" in "In this altitude..."

Replaced.

line 280:    Add a "degree sign" to "(30 latitude band)"

Added.

line 387:    Replace "is" with "occur" in "These differences is probably..."

Replaced.

Figure A1:    In the caption, InvSURF refers to one of the inversion experiments described in section 2.4. But the first time this figure is referred to is in section 2.1 (line 87), when the inversion experiments are not mentioned yet. So suggest to update this caption with a reference to section 2.4, e.g.: "… ground-based surface observations (dots and x-marks) assimilated in the InvSURF inversion experiment (see section 2.4)…"

The phrase in caption is modified accordingly.

Figure A4/A5:   replace TOM with ATom in the legend of both figures

Fixed.

Figure A7:    the dates on the x-axis overlap

Fixed.